# Incremental Learning of Sparse Attention Patterns in Transformers

## Abstract

This paper studies simple transformers on a high-order Markov chain, where the model must incorporate knowledge from multiple past positions, each with different statistical importance. We show that transformers learn the task incrementally, with each stage induced by the acquisition or copying of information from a subset of positions via a sparse attention pattern. Notably, the learning dynamics transition from competitive, where all heads focus on the statistically most important attention pattern, to cooperative, where different heads specialize in different patterns. We explain these dynamics using a set of simplified differential equations, which characterize the stage-wise learning process and analyze the training trajectories. As transformers progress through these stages, they climb a complexity ladder defined via simpler misspecified hypothesis classes until reaching the full model class. Overall, our work provides theoretical explanations for how transformers learn in stages even without an explicit curriculum and provides insights into the emergence of complex behaviors and generalization, with relevance to applications such as natural language processing and algorithmic reasoning.

## 1 Introduction

Knowledge is often compositional and hierarchical in nature. As such, understanding complex concepts often requires an *incremental* approach, where simpler concepts are learned first and then combined to form more complex ideas. Such incremental approaches are crucial for various cognitive tasks, including language comprehension, problem-solving, and decision-making in humans and has been recapitulated in machine learning in various settings (Saxe et al., 2019). In particular, language, is inherently hierarchical, e.g., understanding a sentence requires understanding the meanings of individual words, phrases, and their structure. Consequentially, there has been interest in understanding *incremental* learning behavior of transformers in sequential tasks (Abbe et al., 2023b; Edelman et al., 2024), particularly in how they build upon previously learned information to understand and generate language (Chen et al., 2024a).

The elementary computational operation that is needed to compose information is *copying*, which is used to duplicate data and then perform downstream computations. In language, copying is essential for tasks such as text generation, where the model must replicate certain phrases or structures from the input to produce coherent and contextually relevant output (Olsson et al., 2022), and, as a means to aggregate information from multiple parts of a text to form a comprehensive understanding. Copying is also a fundamental operation in algorithmic reasoning, where it is often necessary to duplicate intermediate results to perform further computations. Transformers implement this operation across different positions via sparse attention patterns which pushes their parameters to diverge. Therefore, the dynamics of how these circuits are established and its implications on reasoning, generalization and emergence are crucial to grasp the inner workings of transformers.

In this paper, we study single-block decoder-based transformers and the formation of sparse attention circuits during training. Simplest such circuit is the "copying" circuit that focused on exactly one position. It is a subcircuit of well-known *induction heads* in transformers (Elhage et al., 2021; Olsson et al., 2022). Sparse attention circuits are the building blocks that allow models to duplicate information from one part of the input to another, enabling the integration of information across multiple positions. We show that they are learned incrementally, with the model first acquiring the ability to copy from the most statistically important pattern, as they provide the most significant

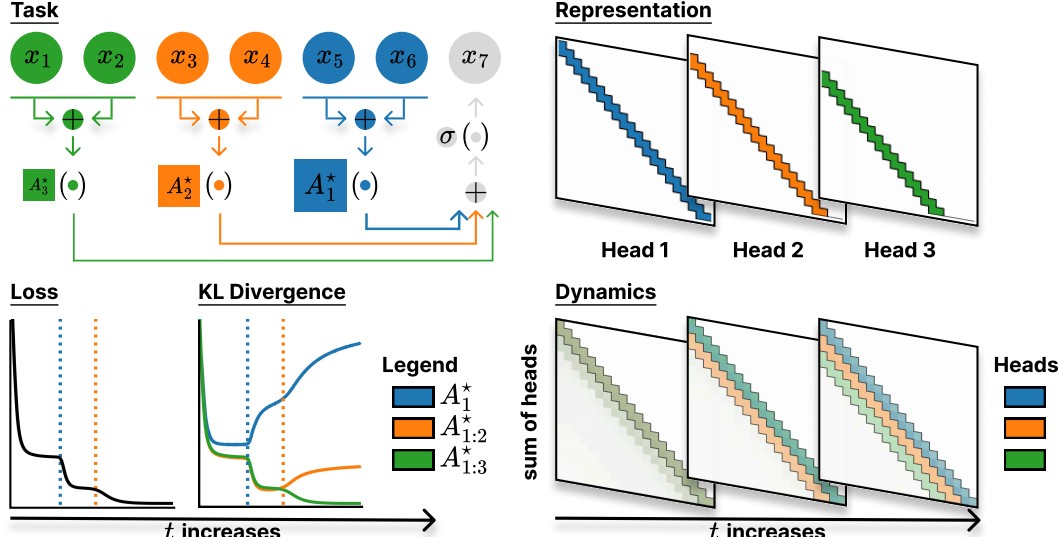

Figure 1: (Top left) The task is based on a high-order Markov chain, where the next token depends on multiple past tokens with different importance weights. The context is divided into different groups of positions, each aggregated and processed by an associated feature matrix $A_k^\star$ of various importance which is represented by the size of the feature matrix. (Top right) An idealized representation of the task in a multi-head single-layer attention. Each head represents an individual sparse attention pattern required to solve the task. (Bottom left) Transformers learn the task incrementally, with each stage corresponding to the acquisition of a sparse attention pattern which is indicated by the KL divergence between predictors $A_{1:i}^\star$ that only depends a subset of relevant positions as defined in Equation (3) and the transformer. (Bottom right) The learning dynamics transition from competitive, where all heads focus on the statistically most important pattern (indicated by high combined attention on the main diagonal), to cooperative, where different heads specialize in different patterns.

improvement in prediction accuracy, and then progressively learning the less important patterns. Interestingly, we observe an initial dynamics where all heads compete to learn the most important pattern, followed by a transition to a cooperative phase where different heads specialize in different patterns. We explain these dynamics using a set of simplified differential equations, after simplifications to the architecture and the task. This leads to connections to tensor factorization which is a well-studied problem (Arora et al., 2019; Razin et al., 2021; Li et al., 2021; Jin et al., 2023).

Our main contributions are as follows:

- We establish the simplest setting for positional incremental learning in transformers. In particular, we isolate the importance of sparse attention patterns as the driving force for incremental learning in transformers, requiring only a single self-attention layer compared to more intricate in-context learning settings such as (Edelman et al., 2024).

- We show that the learning dynamics transition from competitive, where all heads focus on the statistically most important positions, to cooperative, where different heads specialize in different positions. We prove a convergence result that explains the behavior of the first competitive phase as a coupled dynamics induced by the symmetry of the initialization. Additionally, we provide explanations on how the collaborative phase arises after the competitive phase.

- We run studies to understand the impact of the incremental training dynamics on generalization. Depending on the size of the training set, models have different attention patterns, e.g., with a smaller training set, the model learns to copy only from the most important positions. This suggests that there is a regularization induced by the training trajectories, where transformers are pushed to be misspecified depending on the size of the training set. With early stopping, this may result in sample complexity benefits in low-data regimes.

## 2 STAGE-WISE FORMATION OF SPARSE ATTENTION PATTERNS

In this section, we describe the data generation process, how transformers can solve it and the experimental evidence towards incremental learning of sparse attention patterns in transformers.

### 2.1 MARKOV CHAINS WITH IMPORTANCE STRUCTURE

We consider a classification task that is based on a discrete Markov chain of order $w$ with states in a dictionary $\mathcal{D}$ with $|\mathcal{D}| = d$. We treat each element of this dictionary as a one-hot vector in $\mathbb{R}^d$. The sequences are generated as follows:

$$x_{-w+1}, \ldots, x_0 \overset{i.i.d.}{\sim} \mathcal{D}, \quad \text{and for all } t \in [T], \quad x_t \sim \text{softmax}\left(\sum_{k=1}^{h} A_k^\star \sum_{i \in I(k)} \alpha_i x_{t-i}\right), \quad (1)$$

where $A_k^\star \in \mathbb{R}^{d \times d}$ are fixed feature matrices, $I(k)$ are disjoint sets that partition $\{0, \ldots, w-1\}$ and $\alpha_i$ are importance weights which verify $\sum_{i \in I(k)} \alpha_i = 1$ for all $k \in [h]$. This task is simple yet non-trivial and captures some features relevant to the practice: (i) it is sequential, requiring the model to integrate information from past positions, (ii) it has a positional structure, as each component of the prediction depends on a subset of the past states, and (iii) different positions can have different importance, as determined by the feature matrices $A_k^\star$ and scalars $\alpha_i$.

As $I(k)$ and $A_k^\star$ can be permuted without changing the data generation process, we assume without loss of generality that $\|A_1^\star\| \geq \|A_2^\star\| \geq \ldots \geq \|A_h^\star\|$ and that $I(1)$ contains the most important positions, i.e., those associated with the largest feature norms. In general, there can be different spectrums of importance within each feature matrix as well as within each $I(k)$ via $\alpha_i$.

One particular choice of interest is to have $I(k)$ to be contiguous blocks of indices that start from the most recent position, i.e., for some $0 < i_1 < i_2 < \ldots < i_{h-1} < w - 1$,

$$I(1) = \{0, \ldots, i_1\}, I(2) = \{i_1 + 1, \ldots, i_2\}, \ldots, I(h) = \{i_{h-1} + 1, \ldots, w - 1\}. \quad (2)$$

This choice is inspired by the natural language where nearby tokens that complete the text into a word or a short phrase should have more statistical correlation over the distant tokens. Notably, when each of the $I(k)$ are singletons, the resulting operation is copying from a particular position and then processing it with a linear feature map. The "copying" operation is of particular interest as it appears in various settings including in-context learning (Brown et al., 2020).

### 2.2 TRANSFORMERS LEARN INCREMENTALLY

We train single-block decoder-based transformers with $h$ heads on sequences sampled as in Equation (1) by minimizing the cross entropy loss over the full sequence except the initial tokens $x_{-w+1}, \ldots, x_0$ that are not sampled from the process. We keep the architecture as close to the standard practice as possible. The architecture and optimization details are provided in Section A.

We sample feature matrices $A_k^\star$ uniformly over orthogonal matrices and then scale with positive scalars $m_k$. These constants are chosen geometrically, i.e., $m_k = m^{h-k} b_0$ where $m > 1$ is the multiplicative constant and $b_0 > 0$ is the base scale. This results in an importance hierarchy in the feature matrices whereas features within the same matrix has the same importance. In particular, $A_1^\star$ has the largest norm and thus contains the most influential features in the process whereas $A_k^\star$ has the smallest norm and thus the least important features. For simplicity, we choose $\alpha_i = 1/|I(I^{-1}(i))|$ where $I^{-1}$ is the inverse of $I$. Lastly, we choose $I(k)$ as in Equation (2) with the same length intervals of size $w/h$. These choices formalize the notion of relative importance between local positions over the distant positions. As $I(1)$ is paired with $A_1^\star$ that has a large norm, the nearby positions influence the next token more that the distant tokens in $I(h)$ that are paired with $A_h^\star$ which has a small norm. The details of all experimental parameters are provided in Section A and additional experiments can be found in Section B.

We observe that the transformers learn the task incrementally, with each stage corresponding to the acquisition of a sparse attention pattern as in Figure 2. All heads start at uniform due to the initialization. Then, they first mainly focus on the positions in $I(1)$ as they are the most statistically

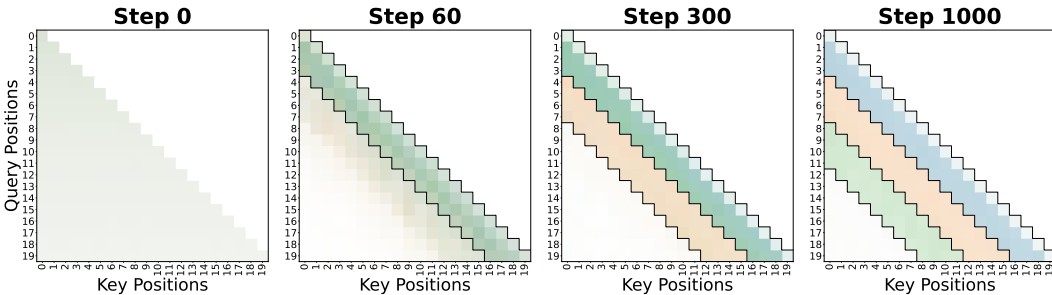

Figure 2: The sum of learned attention patterns for $h = 3, w = 12$ at different stages of training where blue, yellow and green colors correspond to different heads. At $t = 0$, the attention is uniform as the model is randomly initialized. At $t = 60$, all heads learn from the positions in $I(1)$, indicated by the overlapping blue, yellow and green colors, with one head focusing on the positions in $I(2)$ with a small attention. At $t = 300$, a head learns from the positions in $I(2)$ whereas two heads still focus on $I(1)$. At $t = 1000$, the model finally learns to integrate all positions where each head specializes in a different pattern. The main diagonal does not have the same intensity as the other positions as it is learned via the skip connection directly from the input.

important positions. At this stage, the heads compete to learn from these positions, resulting in overlapping attention patterns with some deviations due to the initialization. Later, heads gradually specialize in different patterns, with one head learning from the positions in $I(2)$ while the other finally focusing on $I(3)$.

In order to understand the dynamics in the function space, we train models with different maximum context lengths $c = 4, 8, 12$. When $c = 4$, the model can only access the positions in $I(1)$ and thus learns only from these positions. When $c = 8$, the model can access the positions in $I(1)$ and $I(2)$ and when $c = 12$, the model can access all the relevant positions and can implement the task perfectly. In Figure 3 (right), we plot the Kullback-Leibler (KL) divergence between the predictions of these transformers and the transformer without any context length restriction. We observe that the transformers first approach the model with $c = 4$ and then $c = 8$ before finally reaching the full model with $c = 12$. This indicates that the transformers not only learn the attention patterns but also simultaneously learn the feature matrices associated with these patterns.

Similarly, we study the KL divergence pattern when comparing the predictions of the transformers with restricted context lengths to the ground truths that only depend on the positions in $I(1)$, $I(1) \cup I(2)$ and $I(1) \cup I(2) \cup I(3)$:

$$f_{A_{1:i}^\star}(x_{t-1}, \ldots, x_{t-w}) = \text{softmax}\left(\sum_{k=1}^{i} A_k^\star \sum_{j \in I(k)} \alpha_j x_{t-j}\right). \tag{3}$$

This is plotted in Figure 3 (left) where we see an identical pattern. These are similar to what Edelman et al. (2024) observed for in-context Markov chain where stages are characterized by sub-n-grams.

### 2.3 REPRESENTATION WITH A SIMPLIFIED MULTI-HEAD ATTENTION

Here, we construct a simple representation on a single-layer multi-head attention that solves the task. Let $X \in \mathbb{R}^{d \times (T+w)}$ be the input data matrix with columns $x_{-w+1}, \ldots, x_0, x_1, \ldots, x_T$. We assume that the positional information is encoded using one-hot vectors in $\mathbb{R}^T$ and concatenated to the data as follows:

$$\tilde{X} = \begin{pmatrix} X \\ I_{T+w} \end{pmatrix} \in \mathbb{R}^{(d+T+w) \times (T+w)} .$$

Then, the transformer takes $\tilde{X}$ as input and produces the output $Y \in \mathbb{R}^{d \times T}$ with columns $y_0, \ldots, y_{T-1}$ as follows:

$$y_t = \text{softmax}\left(\sum_{k=1}^{h} V_k \tilde{X} a_t^{(k)}\right), \quad a_t^{(k)} = \text{softmax}\left(\mathcal{M}_{T-t}\left(\tilde{X}^\top K_k^\top Q_k x_t\right)\right),$$

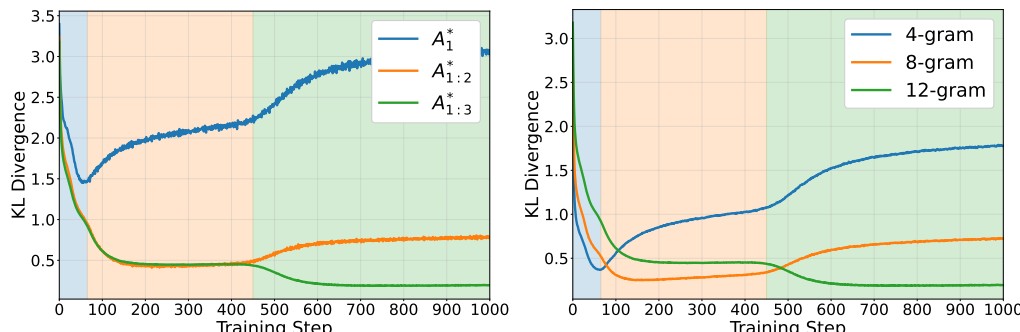

Figure 3: (Left) KL divergence between the ground truths that only depend on the positions in $I(1)$, $I(1) \cup I(2)$ and $I(1) \cup I(2) \cup I(3)$, and the predictions of the transformer with unrestricted context length. (Right) KL divergence between the predictions of the transformers with restricted context lengths $c = 4, 8, 12$ and the transformer without any context length restriction. The transformers learn the task incrementally, with each stage corresponding to the acquisition of information from a subset of positions.

where $Q_k, K_k, V_k \in \mathbb{R}^{(d+T+w) \times (d+T+w)}$ are the query, key and value matrices of the head $k$, respectively and $\mathcal{M}_p$ sets the last $p$ entries to $-\infty$ to apply causal masking.

For head $k$, we set the value matrix $V_k = A_k^\star$ and $a_t^{(k)}$ to be a positional-only attention corresponding to $I(k)$ with the following sparse pattern

$$\frac{1}{|I(k)|} \left( \underbrace{0, \ldots, 0}_{t \text{ entries}}, \underbrace{\mathbf{1}_{1 \in I(k)}, \mathbf{1}_{1 \in I(k)}, \ldots, \mathbf{1}_{1 \in I(k)}}_{w \text{ entries}}, \underbrace{0, \ldots, 0}_{(T-t) \text{ entries}} \right) .$$

Here, the first $t$ entries correspond to the irrelevant tokens in the context and the last $(T - t)$ entries are zeroed out due to the causal masking. Among the relevant tokens in the intermediate $w$ positions, the attention focuses on the indices in $I(k)$ as they can be processed altogether with the same feature matrix $V_k = A_k^\star$. As the target patterns are sparse, the parameters of the attention need to diverge to infinity to exactly learn this operation. In practice, we expect finite values that approximate these sparse attention patterns. These attention patterns can be learned based on the positional information:

$$K_k^\top Q_k = \lambda \sum_{i \in I(k)} \sum_{p=w}^{T+w} e_{d+p-i} e_{d+p}^\top ,$$

where $\lambda > 0$ is a scaling constant and $e_i$ is the $i$-th standard basis vector in $\mathbb{R}^{d+T}$. As $\lambda \to \infty$, the attention scores converge to the desired sparse pattern.

Note that this construction is not unique as there are many $Q_k$ and $K_k$ that can realize the same attention pattern. In particular, there is a symmetry where $(Q_k, K_k)$ can be replaced with $\left( M^{-1} Q_k, M^\top K_k \right)$ for any invertible matrix $M$ without changing the attention scores. Moreover, as there are $h$ heads to learn, the construction has a permutation symmetry among the heads. The permutation symmetry is key in understanding the learning dynamics, as we show in Section 3.2.

## 2.4 ABLATION STUDIES

In order to isolate the essential components that drive the incremental learning behavior, we simplify the architecture by removing some components. First, we remove any components such as layer normalization and residual connections that are not present in the idealized construction in Section 2.3. Then, we reduce the product $K_k^\top Q_k$ to a single matrix $A_k$ as there is a symmetry between $K_k$ and $Q_k$. All of these changes individually or combined do not alter the incremental learning behavior. We plot the learning behavior of this simplified model in Figure 1.

We also perform ablation studies with this minimal architecture. We first vary the initialization scale of the attention matrices $A_k$ and set value matrices to be zero. While initializing $A_k$, we use uniform

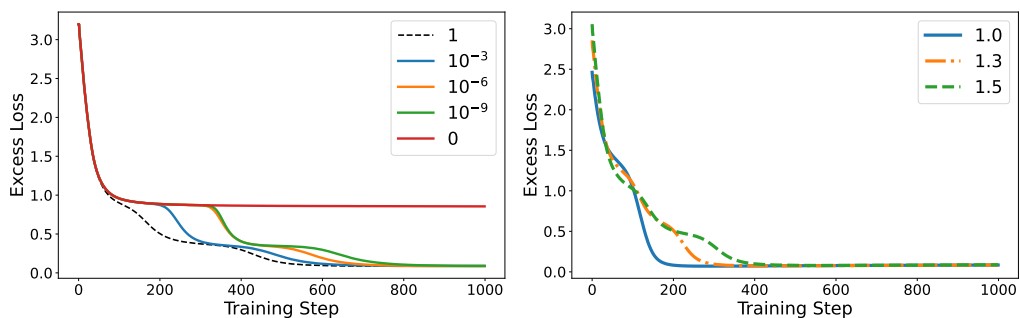

Figure 4: (Left) Excess loss of the minimal architecture with different initialization scales. (Right) Excess loss of the minimal architecture with different multiplicative constants $m$ that determine the importance hierarchy.

distribution over $[-u, u]$ where $u$ is the initialization scale. Figure 4 (left) shows that the speed of incremental learning is affected by the initialization scale, with smaller scales resulting in slower learning. At the extreme $u = 0$, we observe that the model only learns a single pattern and does not progress further. This is because of the symmetry between the heads, which requires a small perturbation to break.

We also vary the multiplicative constant $m$ that determines the structure in the data generation process. Figure 4 (right) shows that the number of steps diminish to two for $m = 1$, where there is no importance ordering. Qualitatively, this model first learns a single pattern and then the other two are learned simultaneously. For $m = 1.3$ and $m = 1.5$, we still observe three distinct stages, but the stages are intertwined for $m = 1.3$ where bumps in the loss landscape are less pronounced.

## 2.5 DATASET SIZE AND GENERALIZATION

Lastly, we study the effect of the dataset size on the incremental learning behavior. As we decrease the dataset size and cross some critical thresholds, we observe that the number of stages that occur in training decreases, as seen in Figure 5 (left). Figure 5 (right) plots the KL divergence between the predictions of the model with different context lengths and the trained transformer. The trend is similar to the one observed in Figure 3 but with different number of bumps for each dataset size.

This points towards a beneficial regularization from the training trajectory which leads to misspecified models, i.e., models that are not able to learn the task perfectly as they have a shorter context length. Yüksel et al. (2025) argue that such misspecification can be beneficial in low-data regimes, making learning statistically feasible. Notably, transformers with early stopping seem to select the misspecification length automatically, hinting at potential sample complexity gains in these settings.

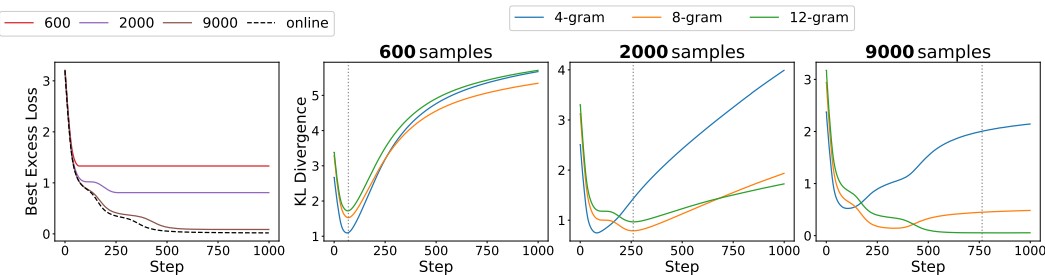

Figure 5: The impact of the dataset size on the incremental learning behavior. (Left) The best validation loss as a function of the dataset size. (Right) The KL divergence between the predictions of the model with different context lengths and the trained transformer. Dashed lines indicate the first step that obtains the best excess loss.

## 3 Training Dynamics on Regression Variant

In this section, we study the regression variant of the classification task in Section 2.1 in Section 2.4. We study the training dynamics in this problem by analyzing the gradient flow dynamics of the loss.

### 3.1 The Regression Model

Consider the following regression task associated to any distribution $\mathcal{P}_X$ and $\mathcal{P}_\xi$

$$(x_1, \ldots, x_T) \sim \mathcal{P}_X, \xi \sim \mathcal{P}_\xi, \quad \text{with} \quad y^\star(X) = \sum_{k=1}^{h} A_k^\star X s_k^\star + \xi, \quad (4)$$

where $s_k^\star \in \mathbb{R}^T$ is the vector with entries $\alpha_i$ for $i \in I(k)$ and zero otherwise. For this section, we set $|I(k)| = 1$ for all $k$ for simplicity. Let $m_k^\star = \|A_k^\star\|_F, V_k^\star = \dfrac{A_k^\star}{m_k^\star}$ for all $k \in [h]$ with $m_1^\star > m_2^\star > \ldots > m_h^\star$ without loss of generality.

We make some assumptions regarding the distributions $\mathcal{P}_X, \mathcal{P}_\xi$ and the feature matrices.

**Assumption 1.** *The noise is zero-mean, i.e., $\mathbb{E}[\xi] = 0$ and the data is normalized, i.e.,*

$$\forall i, j \in [T], \quad \mathbb{E}\left[x_i x_j^\top\right] = \mathbf{1}_{i=j} I_d.$$

**Assumption 2.** *The feature matrices are orthogonal, i.e.,*

$$\forall i, j \in [h], \quad \langle V_i^\star, V_j^\star \rangle = \text{Tr}\left((V_i^\star)^\top V_j^\star\right) = \mathbf{1}_{i=j}.$$

We use the minimal architecture obtained in Section 2.4 with the following modifications. The attention scores are computed only via the inner product of position vectors instead of the concatenated position and data vectors. As the problem is a regression task on the final token, we only need the last row of the matrix $Q_k$ which we denote by $q_k \in \mathbb{R}^T$. Then, the resulting model is as follows:

$$y_\theta(X) = \sum_{k=1}^{h} V_k X s_k, \quad \text{with} \quad s_k = \text{softmax}(q_k),$$

where $\theta = (V_1, \ldots, V_h, q_1, \ldots, q_h)$ are the learnable parameters of the model. We set the loss to the mean square loss:

$$\mathcal{L}(\theta) = \frac{1}{2} \mathbb{E}_{x_1, \ldots, x_T, \xi}\left[\|y_\theta(X) - y^\star(X, \xi)\|^2\right]. \quad (5)$$

We study the gradient flow dynamics of the population loss in Equation (5), i.e., we consider the continuous-time limit of gradient descent with infinitesimal step size.

**Tensor Notation.** We construct tensors that are sum of outer products of matrices and vectors, i.e., $\mathbf{M} = \sum_{k=1}^{h} B_k \otimes v_k$ where $B_k \in \mathbb{R}^{d \times d}$ and $v_k \in \mathbb{R}^T$. The product $X^\top \mathbf{M}$ denotes $X^\top \mathbf{M} = \sum_{k=1}^{h} \langle B_k, X \rangle v_k$ whereas the product $\mathbf{M} v$ denotes $\mathbf{M} v = \sum_{k=1}^{h} B_k \langle v_k, v \rangle$. The inner product between two tensors $\mathbf{M} = \sum_{k=1}^{h} B_k \otimes v_k$ and $\mathbf{N} = \sum_{k=1}^{h} B_k' \otimes v_k'$ is denoted by $\langle \mathbf{M}, \mathbf{N} \rangle = \sum_{k=1}^{h} \langle B_k, B_k' \rangle \langle v_k, v_k' \rangle$. The Frobenius norm of a tensor $\mathbf{M}$ is given by $\|\mathbf{M}\|_F = \sqrt{\langle \mathbf{M}, \mathbf{M} \rangle}$.

Proposition 1 reinterprets this dynamics as a gradient flow of a tensor factorization problem.

**Proposition 1.** *The gradient flow dynamics of the loss in Equation (5) is equivalent to a gradient flow on the following loss:*

$$\mathcal{L}(\theta) = \frac{1}{2} \|\mathbf{G} - \mathbf{P}\|_F^2, \quad \text{where} \quad \mathbf{P} = \sum_{k=1}^{h} V_k \otimes s_k \quad \text{and} \quad \mathbf{G} = \sum_{k=1}^{h} m_k^\star \left(V_k^\star \otimes s_k^\star\right).$$

**Attention Reparameterization.** Note that due to the softmax operation, $\sum_i q_i$ is always constant and thus we can restrict $q_k$ to have a zero mean without loss of generality. This implies that, there is a one-to-one correspondence between $q_k$ and $s_k$ in the subspace of zero-mean vectors. Therefore, it is possible to analyze the dynamics in terms of $s_k$ instead of $q_k$:

$$\begin{aligned}
\dot{V}_k &= (\mathbf{G} - \mathbf{P}) s_k, \\
\dot{s}_k &= \Pi(s_k)^2 \left(V_k^\top (\mathbf{G} - \mathbf{P})\right), \quad \text{where} \quad \Pi(s) = \left(\text{diag}(s) - ss^\top\right).
\end{aligned} \quad (6)$$

**Numerical Simulations.** We simulate these differential equations with initialization $V_i = 0$ and $s_i \approx \frac{1}{T} 1_T$. The results recapitulate the incremental learning behavior observed in Figure 2. We present the results in Section B.4.

## 3.2 COUPLED DYNAMICS DESCRIBE THE COMPETITIVE PHASE

We show that the competitive phase of the learning dynamics can be described by the symmetric initialization $s_1(0) = s_k(0), V_1(0) = V_k(0)$ for all $k$. Once the heads are coupled, they coevolve, i.e., $s_k(0) = s(0), V_k(0) = V(0)$ for all $k$. This leads to the following coupled dynamics:

$$\dot{V} = \left( \boldsymbol{G}s - H\|s\|^2 V \right), \quad \dot{s} = \Pi(s)^2 \left( V^\top \boldsymbol{G} - H\|V\|_F^2 s \right).$$

**Theorem 1.** *Assume that the initialization verifies the following for all $k \in [h]$:*

$$\langle V(0), V_1^\star \rangle \geq \langle V(0), V_k^\star \rangle \quad \langle s(0), s_1^\star \rangle \geq \langle s(0), s_k^\star \rangle. \tag{7}$$

*Then, the dynamics of $V$ and $s$ converge to the following fixed point:*

$$V(\infty) = \frac{m_1^\star}{h} V_1^\star, \quad s(\infty) = s_1^\star. \tag{8}$$

Theorem 1 is based on an ordering argument. As long as the initialization verifies the ordering condition in Equation (7), the dynamics of $V$ and $s$ are such that $\dot{V}$ and $\dot{s}$ reinforces the same order. Standalone, Theorem 1 does not explain what happens when the heads do not start with the same initialization. Theorem 2 establishes that when many heads are initialized with a small deviation from the symmetric initialization, the deviation from the symmetric initialization is bounded for a finite time that we can precisely control. Therefore, the initialization chooses the coupling time of different heads after which they might start to diverge.

**Theorem 2.** *Assume that the following holds for some $\epsilon \ll 1$:*

$$\forall k \in [h] : \|V(0) - V_k(0)\|_F \leq \epsilon \quad and \quad \|s(0) - s_k(0)\|_2 \leq \epsilon.$$

*Then, there exists a universal constant $c_1$ such that*

$$\|V_k(t) - V(t)\|_F \leq \epsilon e^{c_1 t} \quad and \quad \|s_k(t) - s(t)\|_2 \leq \epsilon e^{c_1 t}, \quad \forall t \in \left[ 0, \frac{1}{-c_1 \log \epsilon} \right].$$

Lastly, we remark that the initialization in Theorem 1 can be further relaxed to a wider basin of attraction around the symmetric initialization of interest. This follows from a similar argument as in Zucchet et al. (2025) who has studied the escape time from this initialization when $h = 1$.

**Remark 1.** *The initialization of interest is $s_k(0) \approx \frac{1}{T} 1_T$ for all $k \in [h]$ as seen in Figure 2. By expanding the dynamics around this initialization with $V_k \approx 0$ for all $k \in [h]$, we get:*

$$\dot{V}_k(0) \approx \frac{1}{T} \boldsymbol{G} 1_T, \quad \dot{s}_k(0) \approx 0.$$

*Similarly, second-order local approximation shows that $s_k$ has the largest increase towards the direction $s_1^\star$. Therefore, we can quantify a wider basin of attraction for Theorem 1 as all $V_k$ and $s_k$ move towards the initialization space defined by Equation (7).*

## 3.3 COOPERATION AFTER COMPETITION

In order to study the cooperative phase after the initial competitive phase, we consider the dynamics of the loss at various initializations around the fixed point in Equation (8). Consider the following initialization scheme:

$$V_1(0) = \ldots = V_{h-1}(0) \approx \frac{m_1^\star}{h} V_1^\star, \quad V_h(0) \approx \frac{m_1^\star}{h} V_1^\star,$$
$$s_1(0) = \ldots = s_{h-1}(0) = s_1^\star, \quad s_h \approx s_1^\star. \tag{9}$$

Now, the dynamics of $s_1, \ldots, s_{h-1}$ remain constant due to the projection. In addition, $V_1, \ldots, V_{h-1}$ are coupled due to the gradient flow. Therefore, the whole system collapses to the three equations, one for $V$ that describes the ensemble and two $V', s'$ that describes the offshooting head:

$$
\begin{aligned}
\dot{V} &= m_1^\star \|s_1^\star\|^2 V_1^\star - (h-1)\|s_1^\star\|^2 V - \langle s_1^\star, s'\rangle V', \\
\dot{V'} &= \boldsymbol{G}s' - (h-1)\langle s_1^\star, s'\rangle V - \|s'\|^2 V', \\
\dot{s'} &= \Pi(s')^2 \left( V'^\top \boldsymbol{G} - (h-1)\langle V', V\rangle s_1^\star - \|V'\|^2 s'\right).
\end{aligned}
\tag{10}
$$

We have a similar control to Theorem 2 for the dynamics of $V, V'$ and $s'$. Theorem 3 establishes that the deviation from the cooperative system is bounded for a finite time that we can precisely control. This is due to a Lyapunov control argument where the norms of $V$ and $V'$ are bounded.

**Theorem 3.** *Assume that the following holds for some $\epsilon \ll 1$:*

$$
\forall k \in [h-1]: \|V(0)-V_k(0)\|_F \leq \epsilon, \|e_1-s_k(0)\|_2 \leq \epsilon, \quad \text{and} \quad \|V'(0)-V_h(0)\|_F \leq \epsilon, \|s'(0)-s_h(0)\|_2 \leq \epsilon.
$$

*Let $\Delta(t)$ be the deviation from the cooperative system in Equation* (10)*:*

$$
\Delta(t) = \max\{\max_k\{\|V_k(t) - V(t)\|_F, \|s_k(t) - s(t)\|_2\}, \|V_h(t) - V(t)\|_F, \|s_h(t) - s(t)\|_2\}.
$$

*Assuming that $\|s'(t) - s_1^\star\| \geq \delta$ for all $t \in \mathbb{R}$, there exists a universal constant $c_1$ such that:*

$$
\Delta(t) \leq \epsilon e^{c_1 t}, \quad \forall t \in \left[0, \frac{1}{-c_1 \log \epsilon}\right].
$$

The dynamics in Equation (10) with the initialization in Equation (9) is interesting as while $V'$ grows in an orthogonal direction $V_\perp$ to $V_1^\star$, $s'$ is still sparse around $s_1^\star$. This is due to the fact that $\Pi(s') = \mathcal{O}(\eta)$ as $s'$ is close to $s_1^\star$ and there is a scale separation between $\dot{s}'$ and $\dot{V}'$. Therefore, when $V'$ grows along some $V_\perp$, the prediction is pushed to include the unnecessary term, $V_\perp x_t$. However, this is instantly cancelled out by the progression of the ensemble, where $V$ learns to offset this by learning $-V_\perp$. This collaborative behavior is best seen in our plots in Figure 10.

The initialization of Equation (9) ensures that $V$ is close to its optimal value, $V^\star$, which is defined in Lemma 1. In fact, we can derive a precise statement about how far $V$ is from $V^\star$ based on how much weight $s'$ puts on the directions that are orthogonal to $s_1^\star$:

**Lemma 1.** *Let $\Delta(t) = V(t) - V^\star(t)$ where*

$$
V^\star(t) = \frac{1}{H-1}\left(m_1^\star V_1^\star - \langle s_1^\star, s'(t)\rangle V'(t)\right).
$$

*Assuming that $\|s'(t) - s_1^\star\| \geq \delta$ for all $t \in \mathbb{R}$, there exist constants $c_1, c_2$ such that*

$$
\|\Delta(t)\|_F \leq e^{-c_2 t}\|\Delta(0)\|_F + \frac{c_1}{c_2}.
$$

Inspired by Lemma 1 and numerical simulations, we approximate the full dynamics by a two-scale analysis where $V$ is optimized faster, leading to the following dynamics:

$$
\dot{V'} = \boldsymbol{G}_{(1)}s'_{(1)} - \|s'_{(1)}\|^2 V', \quad \dot{s'} = \Pi(s')^2\left(V'^\top \boldsymbol{G}_{(1)} - \|V'\|_F^2 s'_{(1)}\right),
\tag{11}
$$

where we introduce the following notation:

$$
\boldsymbol{G}_{(i)} = \boldsymbol{G} - \sum_{j=1}^i m_j^\star \left(V_j^\star \otimes s_j^\star\right), \quad s_{(i)} = s - \sum_{j=1}^i \langle s_j^\star, s\rangle s_j^\star.
$$

We show that dynamics in Equation (11) convergences to the second positional feature:

**Theorem 4.** *Assume that the initialization verifies the following for all $k \in [2, h]$:*

$$
\langle V'(0), V_2^\star\rangle \geq \langle V'(0), V_k^\star\rangle \quad \langle s'(0), s_2^\star\rangle \geq \langle s'(0), s_k^\star\rangle.
$$

*Further, suppose that $V'(0), s'(0)$ are such that*

$$
\langle V'(0), \boldsymbol{G}_{(1)}s'_{(1)}(0)\rangle > \frac{1}{2}\|V'(0)\|_F^2\|s'_{(1)}\|^2.
\tag{12}
$$

*Then, the dynamics of $V'$ and $s'$ converge to the following fixed point:*

$$
V'(\infty) = V_2^\star, \quad s'(\infty) = s_2^\star.
$$

Theorem 4 is similar in nature to Theorem 1. Once there is an alignment to the second positional feature, the dynamics is such that this is not broken. In Section D.2, we explain how the same approach can be used to analyze offshooting of an arbitrary head $n$, after the system has learned the first $n-1$ features.

## 4 RELATED WORK

**Incremental learning.**  Plateau-like learning curves are a common feature in neural network training. Early analyses, such as Fukumizu & Amari (2000), attributed these behaviors to critical points in supervised learning. Subsequent studies have examined similar dynamics in a variety of simplified settings, including linear networks (Gissin et al., 2020; Saxe et al., 2019; Gidel et al., 2019; Arora et al., 2019; Jacot et al., 2021; Li et al., 2021; Razin et al., 2021; Jiang et al., 2022; Berthier, 2022; Pesme & Flammarion, 2023; Jin et al., 2023; Varre et al., 2023; 2024), ReLU models (Boursier et al., 2022; Abbe et al., 2023a), and simplified transformer architectures (Boix-Adsera et al., 2023). In transformer training, plateaus followed by sudden capability gains (Chen et al., 2024a; Kim et al., 2024) are often observed in regression tasks (Garg et al., 2022; Von Oswald et al., 2023; Ahn et al., 2024), and formal language recognition (Bhattamishra et al., 2023; Akyürek et al., 2024; D'Angelo et al., 2025).

**n-gram models.**  n-gram language models (Jurafsky & Martin, 2009) serve as a toy setting to understand large language models. This perspective has motivated a range of studies: the optimization landscape has been characterized in Makkuva et al. (2024), expressivity over n-gram distributions has been examined in Svete & Cotterell (2024) and sample complexity has been resolved in Yüksel & Flammarion (2025). Learning of variable-order n-grams have been studied by (Zhou et al., 2024) whereas (Deora et al., 2025) consider $n$-grams with different order.  Connections between ICL and the emergence of induction heads (Elhage et al., 2021; Olsson et al., 2022), together with their acquisition via gradient descent (Nichani et al., 2024), are drawn by Bietti et al. (2024). Training dynamics on n-gram prediction tasks have also been shown to progress in stages: intermediate solutions approximate sub-n-grams (Edelman et al., 2024; Chen et al., 2024b), which later are formalized as near-stationary points by Varre et al. (2025). Despite leading to rich phenomenology, n-grams are typically studied without any inherent hierarhical abstractions that are present in natural language (Wu et al., 2022; 2025). We also use a simplified synthetic data to isolate the phenomenon of study.

**Dynamics of attention.**  The dynamics of diagonal attention have been studied by (Abbe et al., 2023b) whereas (Zhang et al., 2025) study linear attention. Closest to our work, (Zucchet et al., 2025) studies a setting that corresponds $h = 1$ in our paper and studies escape time from the initialization $V = 0, s = \frac{1}{T} 1_T$. Their main technique of analysis is local Taylor approximation around this origin whereas our analysis is on the saddle to saddle dynamics that follows after this initial escape when $h > 1$.

## 5 CONCLUSION

In this work, we have provided a simple but rich task in which transformers need to implement multiple sparse attention patterns. We have shown that it captures the essence of position-dependent incremental learning in transformers. The learning dynamics start competitive where all the heads try to learn the most important pattern. We explain this stage via a coupled dynamics of the attention matrices. After this stage, the heads start to collaborate where the offshooting head learns to predict the other patterns. Our results capture the interplay of sparsity of attention patterns and the learning dynamics of transformers. This is crucial for understanding behavior of transformers in real-world tasks such as reasoning and natural language processing.

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

## ORGANIZATION OF THE APPENDIX

The appendix is organized as follows,

- Section A provides the experimental details.

- Section B presents additional experiments.

- Sections C and D provide proofs of the theoretical results.

- Section E discusses how the initialization in our main theorems can be relaxed.

## A   EXPERIMENTAL DETAILS

The full model has a standard single-layer transformer decoder architecture as discussed in Section 2.2. It uses absolute positional encodings with learnable embedding and unembedding matrices and has the configuration shown in Table 3. The minimal model, as described in Section 2.3, removes layer normalization, dropout, residual connections, key and output attention matrices and the MLP layer. It uses one-hot positional encodings and does not have embedding and unembedding matrices. Both the full model and the minimal model are trained with the same optimization hyperparameters listed in Table 2, and the same synthetic data generation process described in Table 1. The main difference in the learning task between the two models is the interval lengths $|I(k)|$ of the Markov process: the full model uses intervals of length 4, while the minimal model uses intervals of length 2, as summarized in Table 4.

We train the $n$-gram models using the same architecture and optimization hyperparameters as the full transformer model but training with windows of size $n$ sliding over the full sequence.

Table 1: Synthetic dataset parameters

| Parameter | Value |
|---|---|
| Heads $h$ | 3 |
| Dictionary size $d$ | 50 |
| Multiplicative constant $m$ | 1.7 |
| Base scale $b_0$ | 10 |
| Sequence length $T$ | 20 |
| Train samples | 9000 |
| Test samples | 3000 |
| Seed | 0 |

Table 2: Optimization hyperparameters

| Parameter | Value |
|---|---|
| Steps | 2000 |
| Batch size | 3000 |
| Gradient clipping | 1.0 |
| Optimizer | AdamW |
| Weight decay | 0.01 |
| Learning rate | 0.003 |
| Scheduler | ReduceLROnPlateau |
| Patience | 10 |
| Factor | 0.5 |

Table 3: Transformer configuration

| Parameter | Value |
|---|---|
| Hidden dimension | 255 |
| Feedforward dimension | 64 |
| Dropout | 0.1 |
| Initialization scale | 1 |
| Number of blocks | 1 |
| Number of heads | 3 |

Table 4: Markov process intervals

| | Full | Minimal |
|---|---|---|
| $w$ | 12 | 6 |
| $I(1)$ | $\{1,2,3,4\}$ | $\{1,2\}$ |
| $I(2)$ | $\{5,6,7,8\}$ | $\{3,4\}$ |
| $I(3)$ | $\{9,10,11,12\}$ | $\{5,6\}$ |

## B ADDITIONAL EXPERIMENTS

We run additional experiments to study incremental learning behavior under different settings. In particular, we study the effect of infinite data versus finite data, different orders of importance with non-uniform interval lengths and the impact of weight decay.

### B.1 INFINITE DATA

Instead of training on a finite dataset of 9000 samples, we train the model with infinite data by sampling a new batch of data at each step. This removes any effect of overfitting in incremental learning. We observe in Figure 6 and Figure 7 that the model still exhibits the same behavior. This experiment is run with the minimal architecture described in Section 2.4.

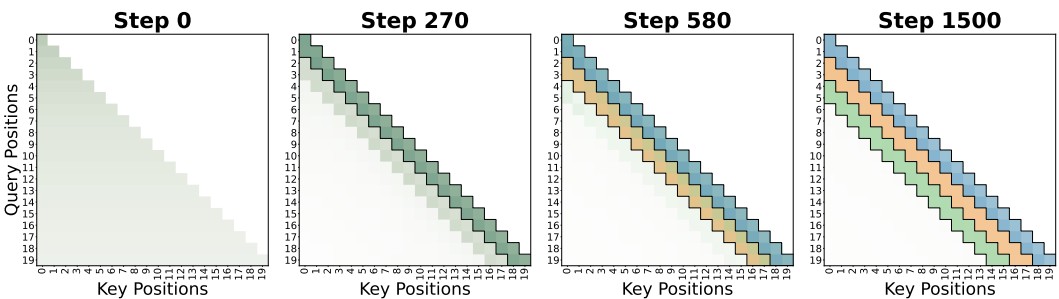

Figure 6: Attention patterns over the training steps with online sampling of data.

### B.2 REVERSE ORDER

We reverse the order of importance of the intervals such that the most important interval is the furthest one. Figure 8 and Figure 9 show the results when $I(3) = \{12,13\}$, $I(2) = \{8,9,10,11\}$ and $I(1) = \{0,1,2,3,4,5,6,7\}$ which reveals the same behavior as the original order. We also note that it is generally easier to observe incremental learning behaviour when the most important interval is the furthest one. This indicates that the learning dynamics is impacted by the sequential structure of the task. This experiment is run with the full architecture described in Section 2.2.

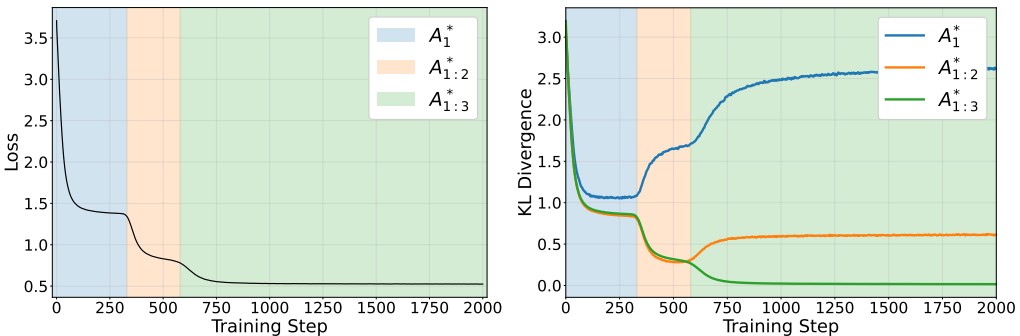

Figure 7: Validation loss and KL divergence over the training steps with online sampling of data.

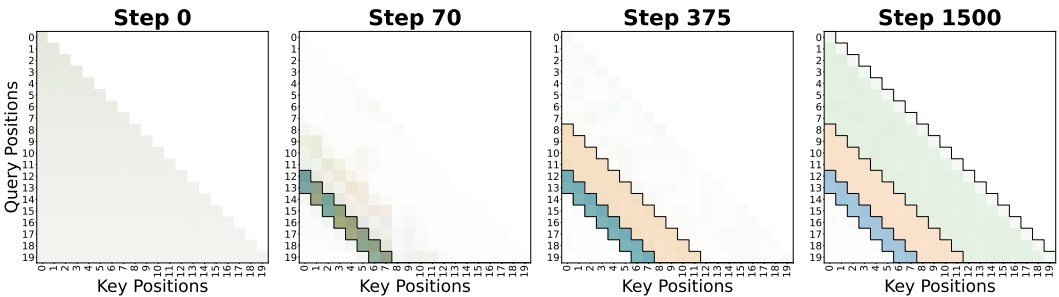

Figure 8: Attention patterns over the training steps with reversed order of importance and varying interval lengths.

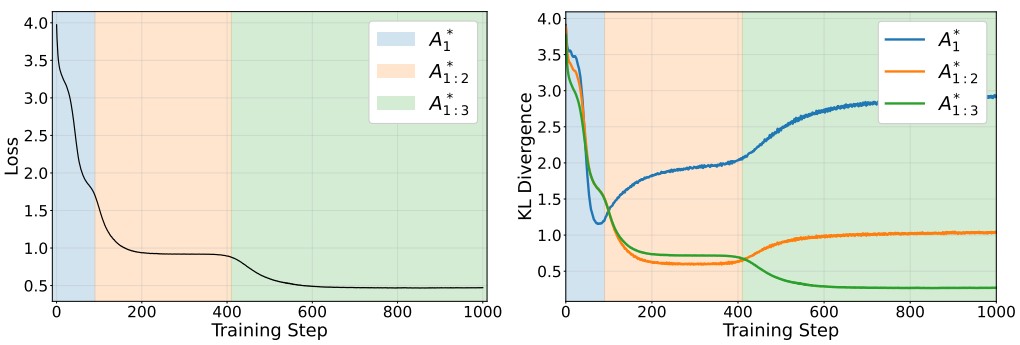

Figure 9: Validation loss and KL divergence over the training steps with reversed order of importance and varying interval lengths.

### B.3 WEIGHT DECAY

We also study the impact of weight decay on the learning dynamics. We observe almost no difference in the learning dynamics when weight decay is not applied so we do not report the results.

### B.4 SIMULATIONS

We present numerical simulations of the gradient flow dynamics of the loss in Equation (5) with the following parameters: $d = 50$, $T = 40$, $h = 3$, $|I(k)| = 1$ for all $k \in [h]$, $m = 1.7$, $\lambda = 0$. We initialize the value parameters $V_i$ to 0 and the attention patterns $s_i$ to $\frac{1}{T}\mathbf{1}_T + \epsilon_i$ where $\epsilon_i$ are sampled from Gaussian distribution with zero-mean and $\epsilon I_T$ covariance with $\epsilon = 10^{-6}$. Figure 10 shows the evolution of the attention patterns $s_k$, the value parameters $V_k$ and the loss over time.

The results aligns with the transformer experiments in Section 2.2. Similar to the transformer experiments, the heads first learn from the position (1) and then the position (2) and finally the position (3). The time scales of these stages are clearly separated where the first stage is the fastest and the third stage is the slowest. Notably, at first, all heads tries to learn from the position (1) as it is related to the most important feature. After this competition phase, the heads start to learn from the position (2) and then the position (3) where they specialize in different patterns. Here, they cooperate to learn from the position (3). In particular, the first head offsets feature (3) as the third head's residual attention on the first position results in a cross term.

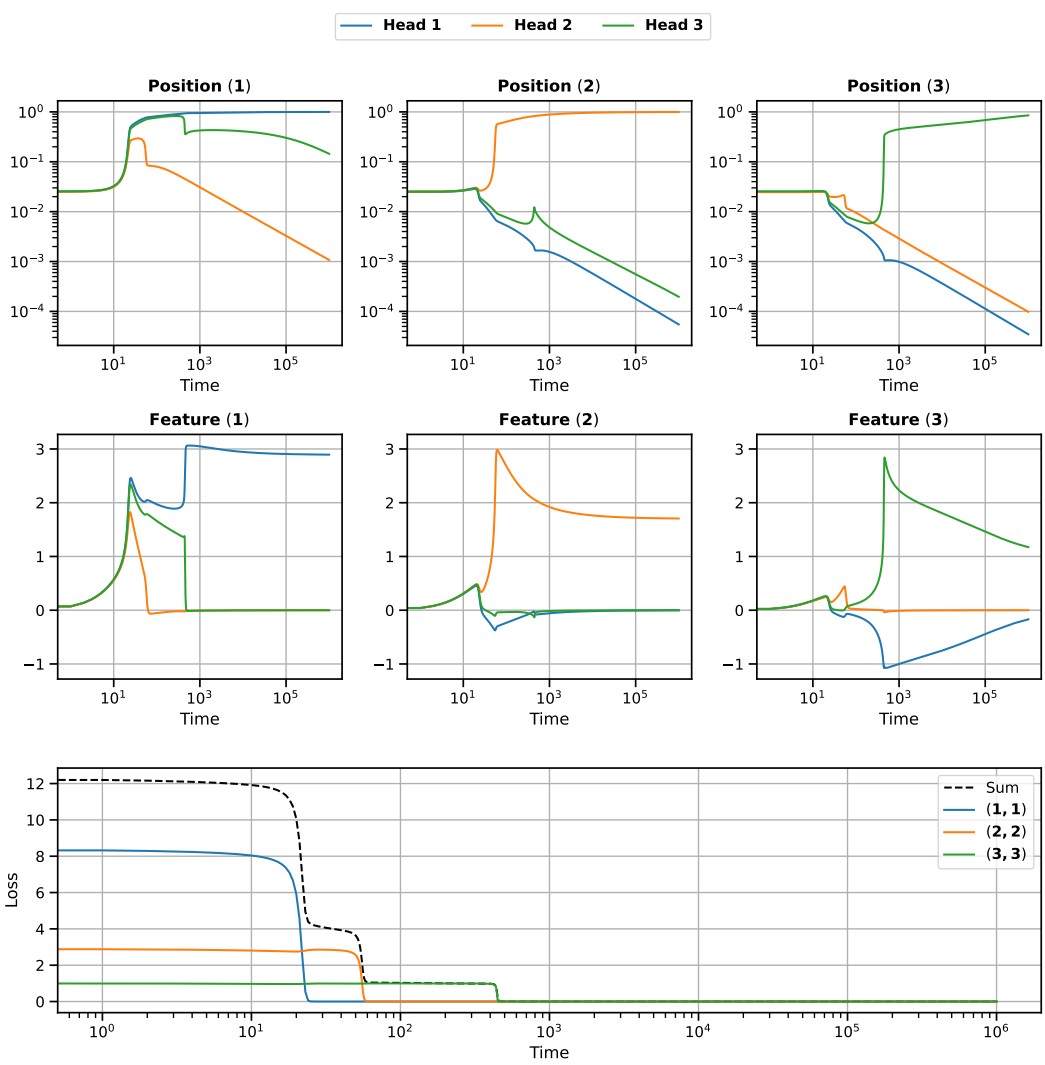

Figure 10: (Top) The evolution of the attention patterns $s_k$ over time. (Middle) The evolution of the value parameters $V_k$ over time. We only plot the relevant coordinates of $s_k$ and $V_k$ for clarity. (Bottom) The evolution of the loss over time. We decompose the loss into the (feature, position) contributions which are plotted in the color of the heads that learn these contributions.

### B.5    TWO-BLOCK TRANSFORMERS

We train 2-block minimal and full transformers with the same configuration as in Section A but adjusting the learning rate and number of training examples. Figures 11 and 12 shows that the incremental learning behavior is similar to the 1-block case. We observe that the first region corresponding to first feature matrix is less pronounced.

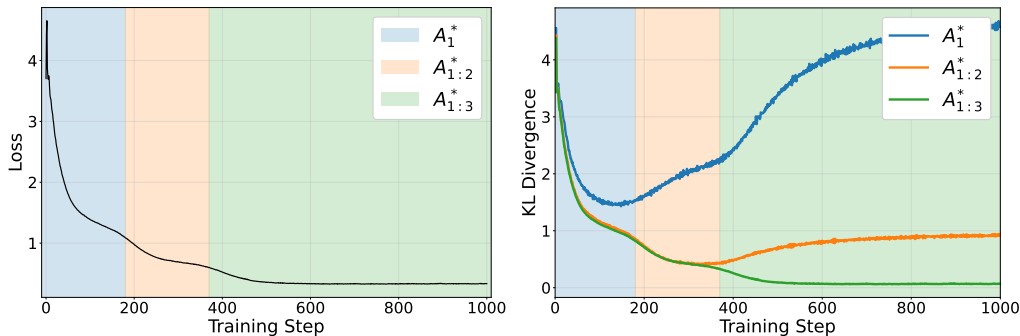

Figure 11: Validation loss and KL divergence over the training steps for a 2-layer minimal transformer.

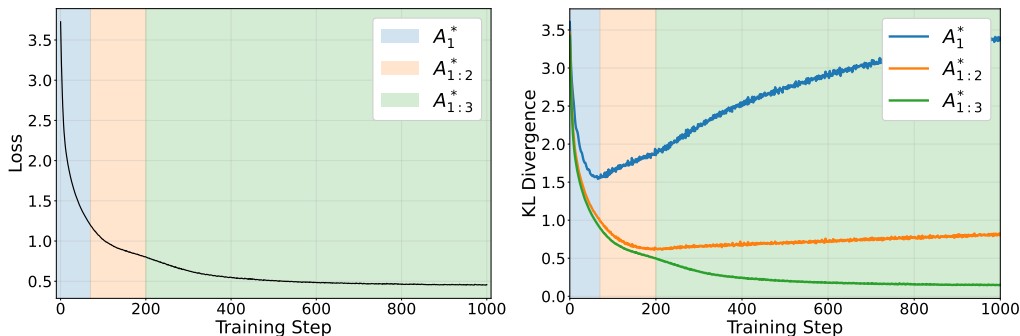

Figure 12: Validation loss and KL divergence over the training steps for a 2-layer full transformer.

## B.6 Non-uniform $\alpha$ values

We run experiments with $\alpha = [0.7, 0.3]$ in Figure 13 and observe that the model still exhibits incremental learning. In Figure 14, we observe the checkered pattern where heads focus more attention on the position with the highest $\alpha$ value.

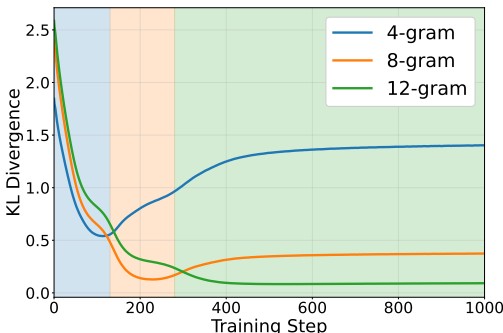

Figure 13: KL divergence over the training steps with non-uniform $\alpha$ values.

## B.7 Overlapping Intervals

We run experiments with overlapping intervals where $I(1) = \{5, 6, 7, 8\}$, $I(2) = \{3, 4, 5, 6\}$, and $I(3) = \{1, 2, 3, 4\}$. This is interval lengths of 4 with an overlap or stride of 2. We try learning transformers with three or four heads. We observe in Figure 15 that the model with four heads still exhibits incremental learning behavior. Similar results are observed for the model with three

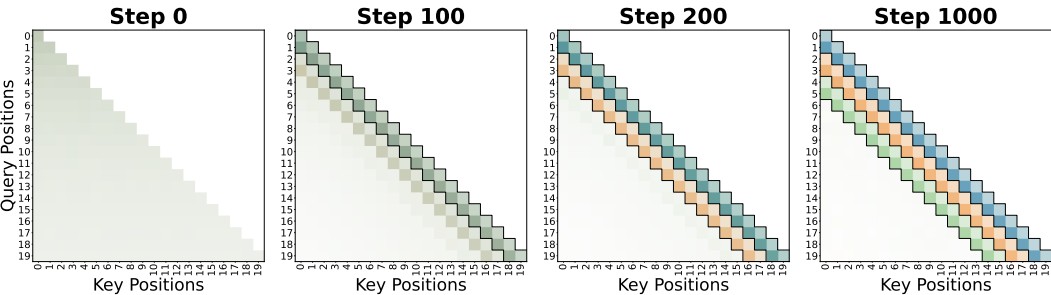

Figure 14: Attention patterns over the training steps with non-uniform $\alpha$ values.

heads and thus omitted. Attention patterns in Figures 16 and 17 reveal the different ordering of learnings for three and four heads. When the intervals are overlapping, it is unclear which positions are statistically the most significant and transformers may follow different solutions based on feature matrices.

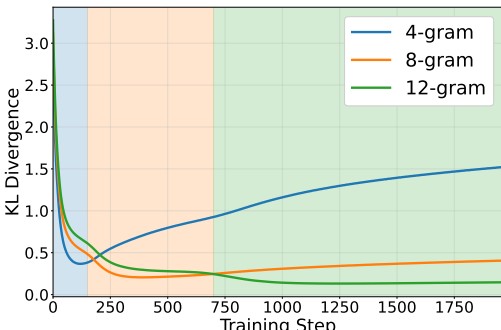

Figure 15: KL divergence over the training steps with intervals of size 4 and overlap of 2 for a transformer model with 4 heads.

### B.8 STOCHASTIC GRADIENT DESCENT (SGD)

We run experiments with SGD optimizer instead of AdamW. We observe in Figure 18 and Figure 19 that the quantitive behavior of incremental learning is same.

## C MISSING PROOFS

Recall that we assume $s_i^\star$ are one-hot in Section 3.1. That is, in the sequel, $\|s_i^\star\|^2 = 1$.

**Proposition 1.** *The gradient flow dynamics of the loss in Equation* (5) *is equivalent to a gradient flow on the following loss:*

$$\mathcal{L}(\theta) = \frac{1}{2}\|\boldsymbol{G} - \boldsymbol{P}\|_F^2\,, \quad \text{where} \quad \boldsymbol{P} = \sum_{k=1}^{h} V_k \otimes s_k \quad \text{and} \quad \boldsymbol{G} = \sum_{k=1}^{h} m_k^\star \left(V_k^\star \otimes s_k^\star\right)\,.$$

*Proof.* We start by some computations. Note that for any vectors $v_1, v_2 \in \mathbb{R}^T$, we have:

$$\mathbb{E}\left[\left(Xv_1\right)\left(Xv_2\right)^\top\right] = \sum_{i=1}^{T}\sum_{j=1}^{T}(v_1)_i(v_2)_j\mathbb{E}\left[x_i x_j^\top\right]$$

$$= \langle v_1, v_2\rangle I_d\,.$$

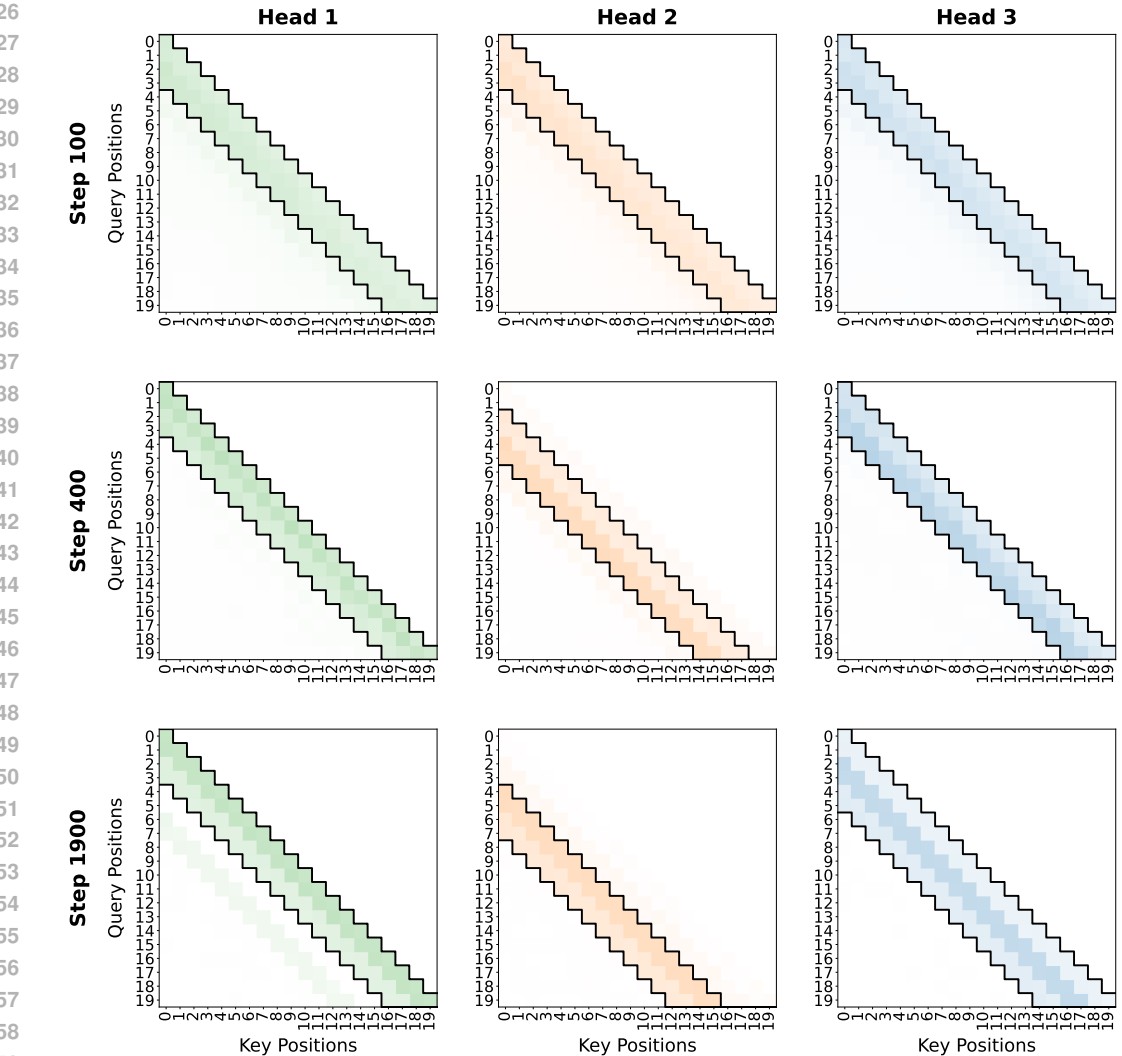

Figure 16: Attention patterns for 3 heads over the training steps with overlapping intervals.

Also, for any vectors $v_1, v_2 \in \mathbb{R}^T$ and any matrix $Q \in \mathbb{R}^{d \times d}$, we have:

$$\mathbb{E}\left[v_1^\top X^\top Q X v_2\right] = \sum_{i=1}^{T} \sum_{j=1}^{T} (v_1)_i (v_2)_j \mathbb{E}\left[x_i^\top Q x_j\right]$$

$$= \sum_{i=1}^{T} \sum_{j=1}^{T} (v_1)_i (v_2)_j \operatorname{Tr}\left(Q \mathbb{E}\left[x_j x_i^\top\right]\right)$$

$$= \langle v_1, v_2 \rangle \operatorname{Tr}(Q).$$

By selecting $v_2 = e_i$ for all $i \in [d]$, we get:

$$\mathbb{E}\left[v_1 X^\top Q X\right] = \operatorname{Tr}(Q) v_1.$$

First, the derivative with respect to $V_i$ is as follows:

$$\frac{\partial \mathcal{L}(\theta)}{\partial V_i} = \mathbb{E}_{X,\xi}\left[\left(f_\theta(X) - f^\star(X, \xi)\right)(X s_i)^\top\right]$$

$$= \sum_{j=1}^{h} V_j \langle s_i, s_j \rangle - \sum_{j=1}^{h} m_j^\star \langle s_i, s_j^\star \rangle V_j^\star.$$

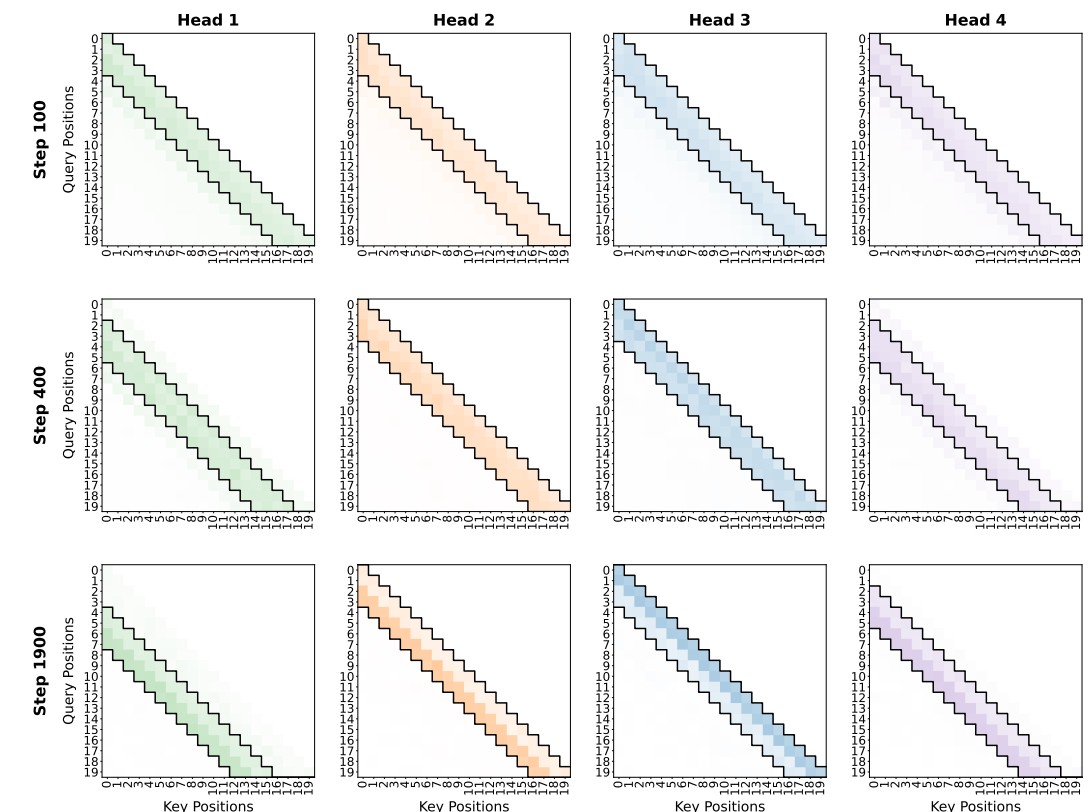

Figure 17: Attention patterns for 4 heads over the training steps with overlapping intervals.

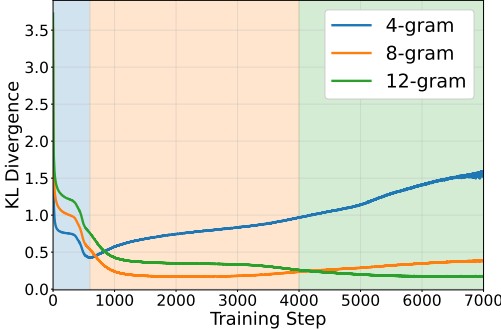

Figure 18: KL divergence over the training steps with SGD optimizer.

Next, the derivative with respect to $q_i$ is as follows:

$$\frac{\partial \mathcal{L}(\theta)}{\partial q_i} = \left(\text{diag}(s_i) - s_i s_i^\top\right) \mathbb{E}_{X,\xi}\left[X^\top V_i^\top \left(f_\theta(X) - f^\star(X,\xi)\right)\right]$$

$$= \left(\text{diag}(s_i) - s_i s_i^\top\right)\left(\sum_{j=1}^h \langle V_i, V_j\rangle s_j - \sum_{j=1}^h m_j^\star \langle V_i, V_j^\star\rangle s_j^\star\right).$$

Then, the gradient flow dynamics is as follows:

$$\dot{V}_i = -\nabla_{V_i}\mathcal{L}(\theta) = (\boldsymbol{G} - \boldsymbol{P})\, s_i$$
$$\dot{q}_i = -\nabla_{q_i}\mathcal{L}(\theta) = \Pi(s_i)\left(V_i^\top\left(\boldsymbol{G} - \boldsymbol{P}\right)\right). \tag{13}$$

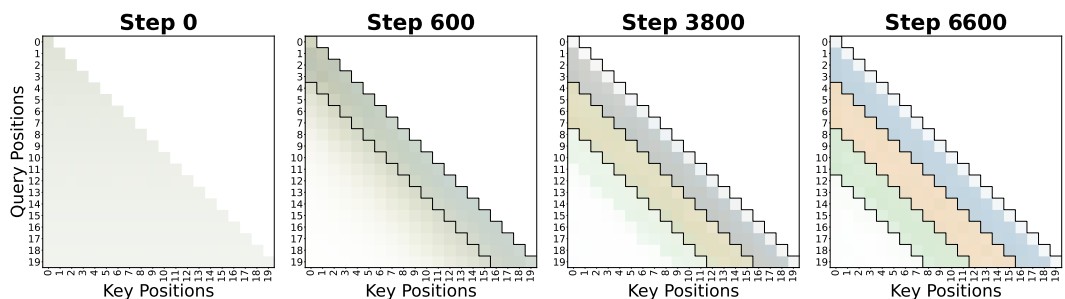

Figure 19: Attention patterns over the training steps with SGD optimizer.

This can be seen as a gradient ascent flow on the following loss:

$$\mathcal{L}(\theta) = \frac{1}{2}\|\boldsymbol{G} - \boldsymbol{P}\|_F^2 \,.$$

$\square$

**Lemma 2.** *Let $s$ be a vector with non-negative entries and $\|s\|_1 = 1$. Then, the kernel space of $\Pi(s) = \mathrm{diag}(s) - ss^\top$ is*

$$\ker\left(\Pi(s)\right) = \mathrm{span}\left(\{e_j : \langle e_j, s\rangle = 0\}\right) \cup \mathrm{span}\left(\sum_{j:\langle e_j,s\rangle>0} e_j\right) \,.$$

*Furthermore, if $\|s\|_1 < 1$,*

$$\ker\left(\Pi(s)\right) = \mathrm{span}\left(\{e_j : \langle e_j, s\rangle = 0\}\right) \,.$$

*Proof.* The proof follows trivially from a rank analysis. $\square$

**Lemma 3.** *Let $s$ be a vector on the simplex that verifies $s_i \geq s_j$ for all $j \in [h]$. Then, for any vector $v$ that verifies $v_i \geq v_j$ for all $j \in [h]$, we have for all $j \in [h]$:*

$$(\Pi(s)v)_i \geq (\Pi(s)v)_j \,.$$

*Proof.* We have the following computations:

$$(\Pi(s)v)_i = s_i\left(v_i - \langle s, v\rangle\right)$$
$$(\Pi(s)v)_j = s_j\left(v_j - \langle s, v\rangle\right) \,.$$

Then, we have:

$$(\Pi(s)v)_i - (\Pi(s)v)_j \geq (s_i - s_j)\left(v_i - \langle s, v\rangle\right) \geq 0 \,.$$

$\square$

**Theorem 1.** *Assume that the initialization verifies the following for all $k \in [h]$:*

$$\langle V(0), V_1^\star\rangle \geq \langle V(0), V_k^\star\rangle \quad \langle s(0), s_1^\star\rangle \geq \langle s(0), s_k^\star\rangle \,. \tag{7}$$

*Then, the dynamics of $V$ and $s$ converge to the following fixed point:*

$$V(\infty) = \frac{m_1^\star}{h}V_1^\star, \quad s(\infty) = s_1^\star \,. \tag{8}$$

*Proof.* Let $\mathcal{R}$ be the following set:

$$\mathcal{R} = \{(V, s) \mid \forall k \in [h], \langle V, V_1^\star - V_k^\star\rangle \geq 0, \langle s, s_1^\star - s_k^\star\rangle \geq 0\} \,.$$

We prove that the flow is forward-invariant on $\mathcal{R}$.

Fix any $j \in [h]$. Let $w_j = \langle V, V_1^\star - V_j^\star \rangle$, $z_j = \langle s, s_1^\star - s_j^\star \rangle$, $r_j = \langle s \odot s, s_1^\star - s_j^\star \rangle$, $t_j = \langle s \odot s \odot s, s_1^\star - s_j^\star \rangle$. The flow of $w_j$ and $z_j$ are as follows:

$$\dot{w}_j = m_1^\star \langle s, s_1^\star \rangle - m_j^\star \langle s, s_j^\star \rangle - H\|s\|^2 w_j \,,$$

$$\dot{z}_j = (s_1^\star - s_j^\star)^\top \Pi(s)^2 \left( V^\top \boldsymbol{G} - H\|V\|_F^2 s \right) \,.$$

Rewriting the derivative of $\dot{z}_j$:

$$\dot{z}_j = \left( (s_1^\star - s_j^\star)^\top \operatorname{diag}(s) - z_j s^\top \right) \Pi(s) \left( V^\top \boldsymbol{G} - H\|V\|_F^2 s \right)$$

$$= (s_1^\star - s_j^\star)^\top \operatorname{diag}(s)^2 \left( V^\top \boldsymbol{G} - H\|V\|_F^2 s \right) - z_j s^\top \operatorname{diag}(s) \left( V^\top \boldsymbol{G} - H\|V\|_F^2 s \right)$$

$$+ \left( \|s\|^2 z_j - r_j \right) \left( V^\top \boldsymbol{G} s - H\|V\|_F^2 \|s\|^2 \right)$$

$$= m_1^\star \langle s_1^\star, s \rangle^2 \|s_1^\star\|^2 \langle V, V_1^\star \rangle - m_j^\star \langle s_j^\star, s \rangle^2 \|s_j^\star\|^2 \langle V, V_j^\star \rangle - H\|V\|_F^2 t_j$$

$$- z_j s^\top \operatorname{diag}(s) \left( V^\top \boldsymbol{G} - H\|V\|_F^2 s \right) + \left( \|s\|^2 z_j - r_j \right) \left( V^\top \boldsymbol{G} s - H\|V\|_F^2 \|s\|^2 \right) \,.$$

On the boundary of $\mathcal{R}$, we have $w_j = 0$ or $z_j = 0$. If $w_j = 0$, then $\dot{w}_j \geq 0$ and if $z_j = 0$, then $r_j = t_j = 0$ and $\dot{z}_j \geq 0$. Therefore, a flow that has started in $\mathcal{R}$ will remain in $\mathcal{R}$ for all time.

Now, consider the following Lyapunov function:

$$\phi(V, s) = \langle V, \boldsymbol{G} s \rangle - \frac{h}{2} \|V\|_F^2 \|s\|^2 \,. \tag{14}$$

The derivative of $\phi(V, s)$ is as follows:

$$\nabla_V \phi(V, s) = \boldsymbol{G} s - H\|s\|^2 V \,,$$

$$\nabla_s \phi(V, s) = V^\top \boldsymbol{G} - H\|V\|_F^2 s \,.$$

Therefore, the time derivative of $\phi$:

$$\dot{\phi}(V, s) = \|\dot{V}\|^2 + \|\Pi(s)\nabla_s \phi(V, s)\|^2 \geq 0 \,.$$

$\phi$ is optimized when $V = \dfrac{\boldsymbol{G} s}{H\|s\|^2}$ which leads to a finite value upper bound on $\phi(V, s)$. Therefore, $\lim_{t \to \infty} \phi(V(t), s(t))$ is finite and the flow converges to a stationary point of $\phi$. That is, the flow converges to a point $(V_\infty, s_\infty)$ that verifies:

$$\boldsymbol{G} s_\infty - H\|s_\infty\|^2 V_\infty = 0 \,, \quad V_\infty^\top \boldsymbol{G} - H\|V_\infty\|_F^2 s_\infty \in \ker(\Pi(s_\infty)) \,. \tag{15}$$

Note that, we have the following equality:

$$(\boldsymbol{G} s_\infty)^\top \boldsymbol{G} = \sum_{j=1}^h m_j^\star \left\langle V_j^\star, \sum_{k=1}^h m_k^\star V_k^\star \langle s_k^\star, s_\infty \rangle \right\rangle s_j^\star = \sum_{j=1}^h (m_j^\star)^2 \langle s_j^\star, s_\infty \rangle s_j^\star \,.$$

Then, the stationary point $(V_\infty, s_\infty)$ verifies

$$\sum_{j=1}^h (m_j^\star)^2 \langle s_j^\star, s_\infty \rangle s_j^\star - h^2 \|s_\infty\|^2 \|V_\infty\|_F^2 \langle s_j^\star, s_\infty \rangle s_j^\star \in \ker(\Pi(s_\infty)) \,. \tag{16}$$

We have proven that $\langle s_1^\star, s_\infty \rangle > 0$ as $\langle s_1^\star, s_\infty \rangle = \max_{k \in [h]} \langle s_k^\star, s_\infty \rangle$. From Lemma 2, $s_1^\star \notin \ker(\Pi(s_\infty))$ as there is at least one index $m \in [T]$ such that $\langle e_m, s_\infty \rangle > 0$ and $\langle e_m, s_1^\star \rangle > 0$. By projecting to the direction $s_1^\star$, Equation (16) implies

$$(m_1^\star)^2 \langle s_1^\star, s_\infty \rangle - h^2 \|s_\infty\|^2 \|V_\infty\|^2 \langle s_1^\star, s_\infty \rangle = 0 \,.$$

However, note that

$$h^2 \|s_\infty\|^2 \|V_\infty\|_F^2 = \frac{\|\boldsymbol{G} s_\infty\|^2}{\|s_\infty\|^2} \leq \max_{\|s\|=1} \|\boldsymbol{G} s\|^2 = (m_1^\star)^2 \,,$$

with equality if and only if $s_\infty = s_1^\star$. Therefore, the flow converges to the stationary point

$$s = s_1^\star \,, \quad V = \frac{m_1^\star}{h} V_1^\star \,.$$

$\square$

**Theorem 2.** *Assume that the following holds for some $\epsilon \ll 1$:*

$$\forall k \in [h] : \|V(0) - V_k(0)\|_F \le \epsilon \quad and \quad \|s(0) - s_k(0)\|_2 \le \epsilon \,.$$

*Then, there exists a universal constant $c_1$ such that*

$$\|V_k(t) - V(t)\|_F \le \epsilon e^{c_1 t} \quad and \quad \|s_k(t) - s(t)\|_2 \le \epsilon e^{c_1 t}, \quad \forall t \in \left[0, \frac{1}{-c_1 \log \epsilon}\right] \,.$$

*Proof.* We write the flow of $V_i$ and $s_i$ in terms of the flow of $V$ and $s$ by new variables:

$$W_i = V_i - V \,, \quad z_i = s_i - s \,.$$

Let $\epsilon$ be the following quantity:

$$\epsilon = \max_{j \in [h]} \max\{\|W_j\|_F, \|z_j\|\} \,.$$

We are interested in the regime where $\epsilon \ll 1$.

Recall that, $\phi(V, s)$ defined in Equation (14) is always non-decreasing. Therefore, $V$ cannot grow larger than $\dfrac{\boldsymbol{G}s}{H\|s\|^2}$ in norm or otherwise $\phi(V, s)$ would decrease. This is the optimal value of $V$ for a particular $s$. Thus, we have a time-independent upper bound $|V| \le \max_s \dfrac{\boldsymbol{G}s}{H\|s\|^2} = \dfrac{m_1^\star}{h}$.

Then, the flow of $W_i$ and $z_i$ is as follows:

$$\dot{W}_i = \boldsymbol{G}z_i - \boldsymbol{P}s_i + H\|s\|^2 V \,,$$
$$\dot{z}_i = \Pi(s_i)^2 \left(V_i^\top (\boldsymbol{G} - \boldsymbol{P})\right) - \Pi(s)^2 \left(V^\top \boldsymbol{G} - H\|V\|^2 s\right) \,.$$

Note that, $\boldsymbol{P}$ can be rewritten as follows:

$$\boldsymbol{P} = \sum_{j=1}^{h} V_j \otimes s_j = HV \otimes s + \left(\sum_{j=1}^{h} W_j\right) \otimes s + V \otimes \left(\sum_{j=1}^{h} z_j\right) + \left(\sum_{j=1}^{h} W_j \otimes z_j\right) \,.$$

This implies that:

$$V^\top \boldsymbol{P} = H\|V\|^2 s + \mathcal{O}\left(\epsilon + \epsilon^2\right) \,, \quad \boldsymbol{P}s = H\|s\|^2 V + \mathcal{O}\left(\epsilon + \epsilon^2\right) \,.$$

We can rewrite the flow of $z_i$ as follows:

$$\dot{z}_i = \left(\Pi(s_i)^2 - \Pi(s)^2\right) \left(V_i^\top (\boldsymbol{G} - \boldsymbol{P})\right) + \Pi(s)^2 \left(W_i^\top \boldsymbol{G} - V_i^\top \boldsymbol{P} + H\|V\|^2 s\right) \,.$$

Therefore, we have:

$$\dot{W}_i = \mathcal{O}(\epsilon), \quad \dot{z}_i = \mathcal{O}(\epsilon) \,.$$

The norm of $W_i$ and $z_i$ are then evolve as follows:

$$\widehat{\|W_i\|} = \frac{\dot{W}_i^\top W_i}{\|W_i\|} \le \|\dot{W}_i\| = \mathcal{O}(\epsilon) \,.$$

We similarly derive that $\|\dot{z}_i\| = \mathcal{O}(\epsilon)$.

This implies that $\epsilon$ verifies the equation:

$$\dot{\epsilon} \le C\epsilon \,, \quad \text{as long as } \epsilon \ll 1 \,,$$

where $C$ is a constant that depends on the problem parameters $H$ and $\boldsymbol{G}$. From the Grönwall's inequality, we have:

$$\epsilon(t) \le \epsilon(0)e^{Ct} \,, \quad \text{as long as } t \in \left[0, \frac{1}{-C \log \epsilon(0)}\right] \,.$$

$\square$

**Theorem 3.** *Assume that the following holds for some $\epsilon \ll 1$:*

$$\forall k \in [h-1]: \|V(0)-V_k(0)\|_F \le \epsilon, \|e_1-s_k(0)\|_2 \le \epsilon, \quad and \quad \|V'(0)-V_h(0)\|_F \le \epsilon, \|s'(0)-s_h(0)\|_2 \le \epsilon.$$

*Let $\Delta(t)$ be the deviation from the cooperative system in Equation* (10)*:*

$$\Delta(t) = \max\{\max_k\{\|V_k(t)-V(t)\|_F, \|s_k(t)-s(t)\|_2\}, \|V_h(t)-V(t)\|_F, \|s_h(t)-s(t)\|_2\}.$$

*Assuming that $\|s'(t)-s_1^\star\| \ge \delta$ for all $t \in \mathbb{R}$, there exists a universal constant $c_1$ such that:*

$$\Delta(t) \le \epsilon e^{c_1 t}, , \quad \forall t \in \left[0, \frac{1}{-c_1 \log \epsilon}\right].$$

*Proof.* We follow the same strategy as in the proof of Theorem 2. The new Lyapunov function is as follows:

$$\phi(V, V', s') = (h-1)m_1^\star \langle V, V_1^\star \rangle - \frac{(h-1)^2}{2}\|V\|_F^2$$

$$- (h-1)\langle s_1^\star, s'\rangle\langle V, V'\rangle + \langle V', \boldsymbol{G}s'\rangle - \frac{1}{2}\|s'\|^2\|V'\|_F^2.$$

We have the following derivatives:

$$\nabla_V \phi(V, V', s') = (h-1)\dot{V},$$
$$\nabla_{V'} \phi(V, V', s') = \dot{V}',$$
$$\nabla_{s'} \phi(V, V', s') = V^\top \boldsymbol{G} - (h-1)\langle V, V'\rangle s_1^\star - \|V'\|^2 s'.$$

By a similar argument, we have that

$$\dot{\phi} = (h-1)\|\dot{V}\|^2 + \|\dot{V}'\|^2 + \|\Pi(s')\dot{s}'\|^2 \ge 0.$$

This indicates that $\phi$ is non-decreasing. By a similar argument to Theorem 2, we establish an upper bound to $\phi$ and consequentially the boundedness of the flow. Then, it is possible to show the noise process grows as $\mathcal{O}(\epsilon)$ where $\epsilon$ is the same quantity as in Theorem 2. $\qquad\square$

# D ANALYSIS OF COOPERATION PHASE

In this section, we extend the analysis in Section 3.3. First, we show convergence of the second head starting with the system in Equation (10). Later, we extend the analysis to any arbitrary phase in the dynamics. We need the following additional notation:

$$\boldsymbol{G}_{(i)} = \boldsymbol{G} - \sum_{j=1}^i m_j^\star \left(V_j^\star \otimes s_j^\star\right), \quad s_{(i)} = s - \sum_{j=1}^i \langle s_j^\star, s\rangle s_j^\star.$$

## D.1 CONVERGENCE OF THE SECOND HEAD

We start with the following initialization scheme:

$$V(0) = \frac{1}{h-1}\left(m_1^\star V_1^\star - \langle s_1^\star, s'(0)\rangle V'(0)\right), \quad V'(0) \sim V(0), \quad s'(0) \sim s_1^\star. \qquad (17)$$

Theorem 1 shows that the initial phase of the dynamics converge to a space close to such an initialization. Here, we note that $\dot{V}(0) = 0$. That is, $V(0)$ is at its optimal value given $V'(0)$ and $s'(0)$. The following lemma shows that $V$ stays close to its optimum through the trajectory:

**Lemma 1.** *Let $\Delta(t) = V(t) - V^\star(t)$ where*

$$V^\star(t) = \frac{1}{H-1}\left(m_1^\star V_1^\star - \langle s_1^\star, s'(t)\rangle V'(t)\right).$$

*Assuming that $\|s'(t)-s_1^\star\| \ge \delta$ for all $t \in \mathbb{R}$, there exist constants $c_1, c_2$ such that*

$$\|\Delta(t)\|_F \le e^{-c_2 t}\|\Delta(0)\|_F + \frac{c_1}{c_2}.$$

*Proof.* Let's compute the derivative of $\Delta$:

$$\dot{\Delta} = -(h-1)\|s_1^\star\|^2 \Delta + \frac{1}{h-1}\langle s_1^\star, s'\rangle \dot{V'} + \frac{1}{h-1}\langle s_1^\star, \dot{s'}\rangle V'.$$

Then, setting $c_2 = \frac{(h-1)}{2}\|s_1^\star\|^2$ and $c(t) = \frac{1}{h-1}\langle s_1^\star, s'(t)\rangle V'(t)$

$$\widehat{\|\dot{\Delta(t)}\|_F^2} = 2\langle \dot{\Delta}(t), \Delta(t)\rangle = -2c_2\|\Delta(t)\|_F^2 + 2\langle \dot{c}(t), \Delta(t)\rangle.$$

We bound the last term as follows:

$$\langle \dot{c}(t), \Delta(t)\rangle \leq \|\dot{c}(t)\|_F \|\Delta(t)\|_F.$$

However, $\dot{c}(t)_F$ is uniformly bounded as in Theorem 3, so we get:

$$\widehat{\|\dot{\Delta(t)}\|_F^2} \leq -2c_2\|\Delta(t)\|_F^2 + 2c_1\|\Delta(t)\|_F.$$

Set $u(t) = \|\Delta(t)\|_F - \frac{c_1}{c_2}$ and rewrite the inequality:

$$\dot{u}(t) \leq -c_2 u(t).$$

By Grönwall's inequality, we have the desired result. $\qquad\square$

Based on Lemma 1 and evidence from our numerical simulations, we approximate the full dynamics by a two-scale analysis where $V$ is optimized faster than $V'$ and $s'$. Compute the gradients of $V', s'$:

$$\dot{V'} = \boldsymbol{G}_{(1)}s'_{(1)} - \|s'_{(1)}\|^2 V', \quad \dot{s'} = \Pi(s')^2\left(V'^\top \boldsymbol{G}_{(1)} - \|V'\|_F^2 s'_{(1)}\right).$$

Expanding $\Pi(s')$, we get

$$\Pi(s') = \Pi(s'_{(1)}) + \langle s_1^\star, s'\rangle^2 s_1^\star (s_1^\star)^\top + \langle s_1^\star, s'\rangle\left(s_1^\star s_{(1)}'^\top + s'_{(1)}(s_1^\star)^\top\right).$$

Since, the $V'^\top \boldsymbol{G}_{(1)} - \|V'\|_F^2 s'_{(1)}$ is perpendicular to the direction $s_1^\star$, we obtain:

$$\dot{s'} = \Pi(s')\left(\Pi(s'_{(1)}) + \langle s_1^\star, s'\rangle s_1^\star s_{(1)}'^\top\right)\left(V'^\top \boldsymbol{G}_{(1)} - \|V'\|_F^2 s'_{(1)}\right).$$

Writing out the update along the direction of $s_1^\star$:

$$\langle s_1^\star, \dot{s'}\rangle = \langle s_1^\star, s'\rangle\|s_1^\star\|^2\left(s_1^\star + \langle s_1^\star, s'\rangle s'_{(1)}\right)^\top\left(\Pi(s'_{(1)}) + \langle s_1^\star, s'\rangle s_1^\star s_{(1)}'^\top\right)\left(V'^\top \boldsymbol{G}_{(1)} - \|V'\|_F^2 s'_{(1)}\right)$$

$$= \langle s_1^\star, s'\rangle^2\|s_1^\star\|^2 s_{(1)}'^\top\left(\Pi(s'_{(1)}) + \|s_1^\star\|^2 I\right)\left(V'^\top \boldsymbol{G}_{(1)} - \|V'\|_F^2 s'_{(1)}\right).$$

The rest of the update follows:

$$\dot{s'_2} = \left(\Pi(s'_{(1)}) + \langle s_1^\star, s'\rangle s'_{(1)}(s_1^\star)^\top\right)\left(\Pi(s'_{(1)}) + \langle s_1^\star, s'\rangle s_1^\star s_{(1)}'^\top\right)\left(V'^\top \boldsymbol{G}_{(1)} - \|V'\|_F^2 s'_{(1)}\right)$$

$$= \left(\Pi(s'_{(1)})^2 + \langle s_1^\star, s'\rangle^2\|s_1^\star\|^2 s'_{(1)}s_{(1)}'^\top\right)\left(V'^\top \boldsymbol{G}_{(1)} - \|V'\|_F^2 s'_{(1)}\right).$$

We are ready to state the main theorem:

**Theorem 4.** *Assume that the initialization verifies the following for all $k \in [2, h]$:*

$$\langle V'(0), V_2^\star\rangle \geq \langle V'(0), V_k^\star\rangle \quad \langle s'(0), s_2^\star\rangle \geq \langle s'(0), s_k^\star\rangle.$$

*Further, suppose that $V'(0), s'(0)$ are such that*

$$\langle V'(0), \boldsymbol{G}_{(1)}s'_{(1)}(0)\rangle > \frac{1}{2}\|V'(0)\|_F^2\|s'_{(1)}\|^2. \tag{12}$$

*Then, the dynamics of $V'$ and $s'$ converge to the following fixed point:*

$$V'(\infty) = V_2^\star, \quad s'(\infty) = s_2^\star.$$

*Proof.* We follow the same strategy as in Theorem 1. Let $\mathcal{R}$ be the following set:

$$\mathcal{R} = \{(V', s') \mid \forall k \in [2, h], \langle V', V_2^\star \rangle \geq \langle V', V_k^\star \rangle \ \text{and} \ \langle s', s_2^\star \rangle \geq \langle s', s_k^\star \rangle \} \ .$$

We prove that the flow is forward-invariant on $\mathcal{R}$.

Fix any $j \in [2, h]$. Let $w_j = \langle V', V_2^\star - V_j^\star \rangle$ and $z_j = \langle s'_{(1)}, s_2^\star - s_j^\star \rangle$. The flow of $w_j$ and $z_j$ are as follows:

$$\dot{w}_j = m_2^\star \langle s'_{(1)}, s_2^\star \rangle - m_j^\star \langle s'_{(1)}, s_j^\star \rangle - \|s'_{(1)}\|^2 w_j \ ,$$

$$\dot{z}_j = \left(s_2^\star - s_j^\star\right)^\top \left(\Pi(s'_{(1)})^2 + \langle s_1^\star, s'\rangle^2 \|s_1^\star\|^2 s'_{(1)} s'^\top_{(1)}\right) \left(V'^\top \boldsymbol{G}_{(1)} - \|V'\|_F^2 s'_{(1)}\right) \ .$$

Rewriting the derivative of $\dot{z}_j$:

$$\dot{z}_j = (s_2^\star - s_j^\star)^\top \Pi(s'_{(1)})^2 \left(V'^\top \boldsymbol{G}_{(1)} - \|V'\|_F^2 s'_{(1)}\right) + cz_j$$

$$= (s_2^\star - s_j^\star)^\top \operatorname{diag}(s'_{(1)})^2 \left(V'^\top \boldsymbol{G}_{(1)} - \|V'\|_F^2 s'_{(1)}\right)$$

$$\quad - (s_2^\star - s_j^\star)^\top \operatorname{diag}(s'_{(1)}) s'_{(1)} s'^\top_{(1)} \left(V'^\top \boldsymbol{G}_{(1)} - \|V'\|_F^2 s'_{(1)}\right) + cz_j$$

$$= (s_2^\star - s_j^\star)^\top \operatorname{diag}(s'_{(1)})^2 \left(V'^\top \boldsymbol{G}_{(1)} - \|V'\|_F^2 s'_{(1)}\right)$$

$$\quad - \left(\langle s'_{(1)}, s_2^\star - s_j^\star \rangle s_2^\star + \langle s'_{(1)}, s_j^\star \rangle \left(s_2^\star - s_j^\star\right)\right)^\top s'_{(1)} s'^\top_{(1)} \left(V'^\top \boldsymbol{G}_{(1)} - \|V'\|_F^2 s'_{(1)}\right) + cz_j$$

$$= m_2^\star \langle s_2^\star, s'_{(1)} \rangle^2 \|s_2^\star\|^2 \langle V, V_2^\star \rangle - m_j^\star \langle s_j^\star, s'_{(1)} \rangle^2 \|s_j^\star\|^2 \langle V, V_j^\star \rangle - \|V\|_F^2 \left(\langle s_2^\star, s'_{(1)} \rangle^3 - \langle s_j^\star, s'_{(1)} \rangle\right) + cz_j \ ,$$

where $c$ is some arbitrary time-dependent function that changes from line to line. On the boundary of $\mathcal{R}$, we have $w_j = 0$ or $z_j = 0$. If $w_j = 0$, then $\dot{w}_j \geq 0$ and if $z_j = 0$, then $\dot{z}_j \geq 0$. Therefore, a flow that has started in $\mathcal{R}$ will remain in $\mathcal{R}$ for all time.

Now, consider the following Lyapunov function:

$$\phi(V', s'_{(1)}) = \langle V', \boldsymbol{G}_{(1)} s'_{(1)} \rangle - \frac{1}{2} \|V'\|_F^2 \|s'_{(1)}\|^2 \ .$$

The derivative of $\phi(V', s'_{(1)})$ is as follows:

$$\nabla_{V'} \phi(V', s'_{(1)}) = \boldsymbol{G}_{(1)} s'_{(1)} - \|s'_{(1)}\|^2 V' \ ,$$

$$\nabla_{s'_{(1)}} \phi(V', s'_{(1)}) = V'^\top \boldsymbol{G}_{(1)} - \|V'\|_F^2 s'_{(1)} \ .$$

Therefore, the time derivative of $\phi$:

$$\dot{\phi} = \|\dot{V}'\|^2 + \|\tilde{\Pi}(s') \nabla_{s'_{(1)}} \phi(V', s')\|^2 \geq 0 \ ,$$

where $\tilde{\Pi}(s')$ is a positive semi-definite matrix that verifies:

$$\tilde{\Pi}(s')^2 = \left(\Pi(s'_{(1)})^2 + \langle s_1^\star, s'\rangle^2 \|s_1^\star\|^2 s'_{(1)} s'^\top_{(1)}\right) \ , \quad \ker(\tilde{\Pi}(s')) \subseteq \ker(\Pi(s'_{(1)})) \ .$$

$\phi$ is optimized when $V' = \dfrac{\boldsymbol{G}_{(1)} s'_{(1)}}{\|s'_{(1)}\|^2}$ which leads to a finite value upper bound on $\phi(V', s'_{(1)})$.

Therefore, $\lim_{t \to \infty} \phi(V'(t), s'_{(1)}(t))$ is finite and the flow converges to a stationary point of $\phi$. That is, the flow converges to a point $(V'_\infty, s'_\infty)$ that verifies:

$$\boldsymbol{G}_{(1)} s'_\infty - H \|s'_\infty\|^2 V'_\infty = 0 \ , \quad V'^\top_\infty \boldsymbol{G}_{(1)} - H \|V'_\infty\|_F^2 s'_\infty \in \ker(\Pi(s'_\infty)) \ .$$

$s'_\infty \neq 0$ as $\phi$ is increasing and satisfy

$$\phi(0) = \phi(V'(0), s'_{(1)}(0)) > 0 \ .$$

Note that, we have the following equality:

$$(\boldsymbol{G}_{(1)} s_\infty)^\top \boldsymbol{G}_{(1)} = \sum_{j=2}^h m_j^\star \left\langle V_j^\star, \sum_{k=2}^h m_k^\star V_k^\star \langle s_k^\star, s'_\infty \rangle \right\rangle s_j^\star = \sum_{j=2}^h (m_j^\star)^2 \langle s_j^\star, s'_\infty \rangle s_j^\star \ .$$

Then, the stationary point $(V'_\infty, s'_\infty)$ verifies

$$\sum_{j=2}^{h} (m_j^\star)^2 \langle s_j^\star, s'_\infty \rangle s_j^\star - h^2 \|s'_\infty\|^2 \|V'_\infty\|_F^2 \langle s_j^\star, s'_\infty \rangle s_j^\star \in \ker(\Pi(s'_\infty)) \,.$$

We have proven that $\langle s_2^\star, s'_\infty \rangle > 0$ as $\langle s_2^\star, s'_\infty \rangle = \max_{k \in [2,h]} \langle s_k^\star, s'_\infty \rangle$. From Lemma 2, $s_2^\star \notin \ker(\Pi(s'_\infty))$. By projecting to the direction $s_2^\star$,

$$(m_2^\star)^2 \langle s_2^\star, s'_\infty \rangle - h^2 \|s'_\infty\|^2 \|V'_\infty\|^2 \langle s_2^\star, s'_\infty \rangle = 0 \,.$$

However, note that

$$h^2 \|s'_\infty\|^2 \|V'_\infty\|_F^2 = \frac{\|\boldsymbol{G}_{(1)} s'_\infty\|^2}{\|s'_\infty\|^2} \leq \max_{\|s\|=1} \|\boldsymbol{G}_{(1)} s\|^2 = (m_2^\star)^2 \,,$$

with equality if and only if $s_\infty = s_2^\star$. Therefore, the flow converges to the stationary point

$$s = s_2^\star, \quad V = m_2^\star V_2^\star \,.$$

$\square$

**Remark 2.** *Equation* (12) *is satisfied by the following initialization*

$$V'(0) = \frac{m_1^\star}{h} V_1^\star + \epsilon V_2^\star, \quad s'(0) = (1-\epsilon)s_1^\star + \epsilon s_2^\star \,,$$

*for small $\epsilon > 0$.*

### D.2 Extension to Higher-order Heads

Similar to Section D.1, we study the offshoot of an arbitrary head $n+1$ after the system has learned the first $n$ features. The features $2, 3, \ldots, n$ are all learned by a single head whereas the ensemble of $h-n$ heads are still on the first feature. This leads to the following dynamics similar to Equation (10):

$$V_1 = V_{n+2} = \ldots = V_h = V, \quad s_1 = s_{n+2} = \ldots = s_h = s_1^\star, \quad s_2 = s_2^\star, \ldots, s_n = s_n^\star \,.$$

We assume an analog of the initialization in Equation (17):

$$V(0) = \frac{1}{h-n} (m_1^\star V_1^\star - \langle s_1^\star, s_{n+1} \rangle V_{n+1}(0)) \,,$$
$$V_i(0) = m_i^\star V_i^\star - \langle s_i^\star, s_{n+1} \rangle V_{n+1}(0), \quad \forall i \in [2, n] \,,$$
$$V_{n+1}(0) \sim V(0), \quad s_{n+1}(0) \sim s_1^\star \,.$$

This leads to a similar dynamics after assuming $V, V_2, \ldots, V_n$ has fast dynamics:

$$\dot{V}_{n+1} = \boldsymbol{G}_{(n)} (s_{n+1})_{(n)} - \|(s_{n+1})_{(n)}\|^2 V_{n+1} \,,$$
$$\dot{s}_{n+1} = \Pi(s_{n+1})^2 \left( V_{n+1}^\top \boldsymbol{G}_{(n)} - \|V_{n+1}\|_F^2 (s_{n+1})_{(n)} \right) \,.$$

The same analysis in Section D.1 leads to the following theorem:

**Theorem 5.** *Assume that the initialization verifies the following for all $k \in [n+1, h]$:*

$$\langle V_{n+1}(0), V_n + 1^\star \rangle \geq \langle V_{n+1}(0), V_k^\star \rangle \quad \langle s_{n+1}(0), s_{n+1}^\star \rangle \geq \langle s_{n+1}(0), s_k^\star \rangle \,.$$

*Further, suppose that $V_{n+1}(0), s_{n+1}(0)$ are such that*

$$\langle V_{n+1}(0), \boldsymbol{G}_{(n)}(s_{n+1})_{(n)}(0) \rangle > \frac{1}{2} \|V_{n+1}(0)\|_F^2 \|(s_{n+1})_{(n)}\|^2 \,.$$

*Then, the dynamics of $V_{n+1}$ and $s_{n+1}$ converge to the following fixed point:*

$$V_{n+1}(\infty) = V_{n+1}^\star, \quad s_{n+1}(\infty) = s_{n+1}^\star \,.$$

# E  EXPANDING THE INITIALIZATION CONDITION

In this section, we explain Remark 1 in detail. As stated, for any initialization around $s_k(0) \approx \frac{1}{T}1_T$ and $V_k \approx 0$, we obtain the following from the first-order Taylor approximation as $\boldsymbol{P} \approx 0$:

$$\dot{V}_k(0) \approx \frac{1}{T}\boldsymbol{G}1_T\,, \quad \dot{s}_k(0) \approx 0\,.$$

Therefore, the heads $V_k(0)$ exhibit a faster dynamics than the attention scores $s_k$. For small timescales $t$, the heads are approximately aligned with the same direction:

$$V_k(t) \approx \frac{t}{T}\boldsymbol{G}1_T\,,$$

which satisfies the initialization condition in Theorem 1 as $m_1^\star \geq m_k^\star$ for any $k \in [h]$. Moreover, the second-order Taylor approximation yields:

$$\ddot{V}_k(0) \approx -\sum_i \dot{V}_i s_i^\top s_k \approx \frac{1}{T^2}\boldsymbol{G}1_T\,,$$

$$\ddot{s}_k(0) \approx \Pi(s_k)\dot{V}_k^\top\left(\boldsymbol{G} - \boldsymbol{P}\right) \approx \frac{1}{T}\pi(s_k)\boldsymbol{G}1_T\,.$$

By, Lemma 3, we can show that $\dot{s}_k(0)$ is such that the component of $s_1^\star$ is the maximal entry. Therefore, we expect $s_k$ to align towards the initialization condition given in Theorem 1 for small timescales $t$:

$$s_k(t) \approx \frac{1}{T^2}\left(I_T - \frac{1}{T}1_T1_T^\top\right)\boldsymbol{G}1_T\,.$$

Similar type of analysis also applies to the initializations of Theorems 4 and 5.

Note that the initialization regimes in Theorems 1, 4 and 5 are not towards a particular point but a large set that verifies some ordering. Coupled with the analysis above, the initialization basin for these theorems can be expanded. This contrasts with analyses that rely on vanishing initialization or a particular limit towards fixed points.

## LLM USAGE STATEMENT

We acknowledge the use of Large Language Models (LLMs) to assist with preparing this manuscript. They were utilized for several tasks: improving the grammar and clarity of the text; generating and editing code snippets; and helping identify relevant literature for our review. All LLM-generated content, including suggestions for literature, was carefully reviewed, verified, and revised by the authors, who take full responsibility for the final paper.

