# OpenReview forum: "Incremental Learning of Sparse Attention Patterns in Transformers"
_ICLR.cc/2026/Conference — Submitted to ICLR 2026_

### Official Review · Reviewer_ShLm · 2025-10-30

**Soundness:** 3
**Presentation:** 2
**Contribution:** 2
**Rating:** 4
**Confidence:** 3

**Summary:**

This paper analyzes how simple Transformers learn on a high-order Markov-chain task where the next token depends on multiple past positions with different statistical importance. The authors observe stage-wise learning: attention first concentrates on the statistically most important positions, then progressively adds less important ones as training proceeds. The dynamics transition from a competitive phase, where all heads chase the same dominant pattern, to a cooperative phase, where different heads specialize in distinct sparse attention patterns. They formalize these behaviors with simplified differential equations for a minimal single-layer multi-head architecture and connect the analysis to tensor factorization. They also study a regression variant and derive gradient-flow dynamics that reproduce the incremental stages. Finally, they report generalization effects tied to dataset size and early stopping, arguing that trajectory-induced misspecification can be beneficial in low-data regimes.

**Strengths:**

### Clear minimal setting that isolates sparse attention

By crafting a high-order Markov process with block-structured lags and importance weights, the paper cleanly isolates when sparse attention should emerge. The construction $I(1),\dots,I(h)$ partitions the past and the norm ordering $\lVert A_1^\star\rVert \ge \cdots \ge \lVert A_h^\star\rVert$ creates a natural “importance ladder,” making stage boundaries interpretable and reproducible. This design sharpens causal attributions between task structure and learned attention patterns.

### Mechanistic account of competitive to cooperative dynamics

A notable empirical finding is that heads initially compete for the same dominant position, then shift to cooperation as specialization emerges across heads. The paper documents this transition with attention maps and with a KL-divergence probe that tracks how many required patterns the model has acquired at any time. This mechanistic narrative clarifies how multi-head attention distributes roles across heads as training advances.

### Simplified differential-equation modeling

The authors complement experiments with a simplified ODE analysis of a minimal single-layer, multi-head model. The resulting coupled dynamics explain the competitive phase under symmetric initialization and illuminate how the cooperative phase arises. This pairing of gradient-flow equations with observations provides a compact theoretical handle on stage-wise emergence.

### Connection to factorization viewpoints

By reducing the problem to structured predictors and aligning the updates with tensor or matrix factorization intuitions, the paper relates attention-pattern acquisition to well-studied decomposition problems. This link helps interpret why heads specialize on different subsets and why importance ordering drives the sequence of stages.

### Generalization insights via training-trajectory regularization

The study on dataset size reports that smaller datasets yield fewer learned stages, effectively selecting shorter effective context and sparser copying behaviors. The authors interpret this as trajectory-induced regularization and show that early stopping can choose a beneficial misspecification level in low-data regimes. This gives a practical takeaway about tuning training length and data scale to trade off bias and variance.

**Weaknesses:**

### Synthetic scope and external validity

All main results are on synthetic high-order Markov data with one-hot vocabularies, blocky lag sets $I(k)$, and fixed feature matrices $A_k^\star$. While this isolates mechanisms, it leaves open how strongly the conclusions transfer to natural language or other real data with heterogeneous, long-range and overlapping dependencies. The paper gestures at relevance, yet concrete tests beyond the synthetic regime are limited.


### Strong simplifying assumptions in the theory

The regression analysis relies on assumptions such as orthogonality of normalized features and model minimality, which streamline the ODEs but narrow applicability. It is unclear how robust the predicted stages and transition timings are when these assumptions are relaxed, for example with correlated features or deeper stacks.


### Limited architectural and optimization diversity

Most analysis targets a single-layer, decoder-style, multi-head setup with symmetric initialization. The paper does not deeply explore how depth, alternative positional encodings, optimizer choices, or regularizers perturb the competitive to cooperative transition. This reduces guidance for practitioners calibrating real training runs.


### Relation to Zucchet et al., (2025) [1] is unclear

I think Zucchet et al., (2025) [1] is a relevant paper, but the relation is not sufficiently discussed in the paper, although the authors cite it in Section 3.2.


[1] Zucchet, Nicolas, et al. "The emergence of sparse attention: impact of data distribution and benefits of repetition." arXiv preprint arXiv:2505.17863 (2025).

**Questions:**

- In your data-generation process $x_t \sim \mathrm{softmax} \Big( \sum_{k=1}^{h} A_k^\star \sum_{i\in I(k)} \alpha_i, x_{t-i}\Big)$ with disjoint lag blocks $I(k)$, how sensitive are the stage boundaries to relaxing the disjointness assumption, for example when $I(k)$ overlap or when importance weights $\alpha_i$ are heavily skewed or noisy?

- Please clarify the relation to Zucchet et al., (2025) [1].

---

> ### Author Response · Authors · 2025-11-21
>
> We thank the reviewer for the compherensive review. Below, we provide our rebuttals to weaknesses and questions by listing the improvements in the manuscript:
>
> > Synthetic scope and external validity
>
> As the reviewer mentions, our setting is idealized to isolate the mechanism of interest. This is a standard approach in the literature for mathematical understanding of transformers [2-3]. $n$-grams are classical models for sequential data and sufficient to exhibit rich phenomenology with transformers [4-6].
>
> We also remark that our data framework is able to address long-range dependencies (by adjusting $w$) or overlapping dependencies (see below).
>
> > In your data-generation process $x_t \mathrm{softmax}\left(\sum_{k=1}^h A_k^\star \sum_{i \in I(k)} \alpha_i x_{t-i}\right)$ with disjoint lag blocks $I(k)$, how sensitive are the stage boundaries to relaxing the disjointness assumption, for example when $I(k)$ overlap or when importance weights $\alpha_i$ are heavily skewed or noisy?
>
> We **added new experiments** in Appendix B.6. and B.7. where we study non-uniform $\alpha$ values and overlapping intervals. For the former, we observe qualitatively the same phenomena as in the uniform case. For the latter, the incremental learning is still present but the ordering of the learning is less clear. We show that students with three and four heads learn different solutions when presented data from a teacher with 3 different intervals.
>
> > Strong simplifying assumptions in the theory
>
> The orthogonality of normalized features and model minimality are standard in the literature [7-8]. Without such assumptions, the dynamics are not tractable.
>
> > Limited architectural and optimization diversity
>
> **Depth**: We added **new experiments to Appendix B.5.** where we show two-block transformers exhibit qualitatively the same behavior. Overall, two-block transformer have less pronounced first stage.
>
> **Positional encodings**: We explore two alternative positional encodings in the paper: learned positional encoding, one-hot positional encoding. We have also experimented with different positional encodings such as sinusodial positional encoding, relative and rotary positional encodings but didn't include them as the findings are qualitatively the same.
>
> **Optimizers**: We added **new experiments to Section B.7** where run SGD instead of AdamW. The results are qualitatively the same.
>
> **Regularizers**: Our experiments with full transformers include dropout and weight decay. These experiments also include layer normalization. In Section B.3., we discuss the effects of weight decay.
>
>
> > Relation to Zucchet et al., (2025) [1] is unclear
>
> [1] studies a setting that corresponds $h=1$ in our paper. They are interested in how fast one escapes the initialization $(V = 0, s = \frac{1}{T} 1_T)$ and the effect of burstiness/repetition on the escape time. The main technique of analysis is local Taylor approximation around this origin. Notably, the origin is not a saddle.
>
> Our theoretical analysis is on the whole dynamics that follows after this initial escape. When $h > 1$, this leads to incremental learning with saddle-to-saddle dynamics. We establish the convergence of the dynamics to the first saddle when heads are coupled. In Appendix D, **we added a new analysis** of convergence to other saddles by assuming certain exiting points from the previous saddles. Here, we need to rely on certain invariances of the trajectories.
> We added this comparison to the related work section.
>
> In Remark 1, we comment how the same Taylor approximation used in [1] can be used to widen the basis of initialization in Theorem 1. We added **a new discussion** that details Remark 1 in Section E.
>
> [2] Marion, Pierre, et al. "Attention layers provably solve single-location regression." arXiv preprint arXiv:2410.01537 (2024).
>
> [3] Duranthon, Odilon, et al. "Statistical advantage of softmax attention: insights from single-location regression." arXiv preprint arXiv:2509.21936 (2025).
>
> [4] Nichani, Eshaan, Alex Damian, and Jason D. Lee. "How transformers learn causal structure with gradient descent." arXiv preprint arXiv:2402.14735 (2024).
>
> [5] Edelman, Ezra, et al. "The evolution of statistical induction heads: In-context learning markov chains." Advances in neural information processing systems 37 (2024): 64273-64311.
>
> [6] D'Angelo, Francesco, Francesco Croce, and Nicolas Flammarion. "Selective induction heads: How transformers select causal structures in context." The Thirteenth International Conference on Learning Representations. 2025.
>
> [7] Boursier, Etienne, Loucas Pillaud-Vivien, and Nicolas Flammarion. "Gradient flow dynamics of shallow relu networks for square loss and orthogonal inputs." Advances in Neural Information Processing Systems 35 (2022): 20105-20118.
>
> [8] Zhang, Y., Singh, A.K., Latham, P.E. &amp; Saxe, A.M. "Training Dynamics of In-Context Learning in Linear Attention." Proceedings of the 42nd International Conference on Machine Learning (2025).

---

> > ### Comment · Reviewer_ShLm · 2025-11-27
> >
> > Thank you for the response.
> >
> > I think the authors addressed many of my concerns through the rebuttal, but I am still not satisfied about the limited scope of the experiments.
> > To give practical implications, I think experiments beyond the synthetic regime are required.
> > Minegishi et al., (2025) is a good example, in which observations in synthetic experiments are also supported by analyzing the internal mechanics of pretrained language models, so the authors can refer to it as a possible experimental design.
> >
> > I keep my score now, but I am still open to discuss further.
> >
> > Thanks.
> >
> > ### References
> >
> > Minegishi, Gouki, et al. "Beyond Induction Heads: In-Context Meta Learning Induces Multi-Phase Circuit Emergence." Forty-second International Conference on Machine Learning.

---

> > > ### Author Response · Authors · 2025-11-27
> > >
> > > We thank the reviewer for the follow-up comment and are pleased that our rebuttal resolved many of the raised issues. We address the remaining concern below in three parts:
> > >
> > > 1. We study the formation of the **copying** circuit which is a sub-circuit of many well-known circuits including **induction heads**. There are numerous works that have observed induction heads and other circuits that rely on copying beyond synthetic regimes [9-12]. All of these works indicate the existence of the copying sub-circuits. Therefore, there is abundant evidence that are equivalent to the one given in Minegishi et al., (2025).
> > >
> > > 2. As we cited earlier [2-6,8], numerous works focused on analyzing simple transformers. The simplicity is important as otherwise various parts of the transformer interact and the phenomena become too complicated to understand. Notably, seminal works [4,5] do not provide any experimental evidence with MLPs, layer normalization or transformers of depth larger than 2. Hence, they also do not satisfy the requirement the reviewer has raised.
> > >
> > > 3. In addition to our empirical investigations of the stage-wise learning of sparse attention patterns, we also develop a principled theory for this phenomenon. Our focus is therefore not on proving its existence in further real-world scenarios, this has already been established as explained, but on providing a rigorous theoretical account. This viewpoint should be factored into the evaluation.
> > >
> > > We are happy to address any further questions or discussions.
> > >
> > > [9] Olsson, Catherine, et al. "In-context learning and induction heads." arXiv preprint arXiv:2209.11895 (2022).
> > >
> > > [10] Wang, Kevin, et al. "Interpretability in the wild: a circuit for indirect object identification in gpt-2 small." arXiv preprint arXiv:2211.00593 (2022).
> > >
> > > [11] Lieberum, Tom, et al. "Does circuit analysis interpretability scale? evidence from multiple choice capabilities in chinchilla." arXiv preprint arXiv:2307.09458 (2023).
> > >
> > > [12] Merullo, Jack, Carsten Eickhoff, and Ellie Pavlick. "Circuit component reuse across tasks in transformer language models." arXiv preprint arXiv:2310.08744 (2023).

---

### Official Review · Reviewer_dRyx · 2025-10-31

**Soundness:** 4
**Presentation:** 2
**Contribution:** 3
**Rating:** 6
**Confidence:** 3

**Summary:**

This paper investigates the learning dynamics of single-block, decoder-only Transformers, focusing on the emergence of sparse attention circuits as network learns to predict sequence tokens that are influenced by a mixture of other token chunks in history. The authors empirically observe that attention heads initially converge on the most statistically dominant pattern and, over time, progressively specialize in less frequent ones.
The paper develops a theoretical framework to explain this phenomenon. Starting from a simplified regression setup, it derives differential equations characterizing the gradient dynamics of multi-head attention learning. The analysis reveals that the process can be viewed as a low-rank tensor regression problem, where the model effectively factorizes a target tensor (the ground truth) into rank-1 components. The dynamics of the network explain that all heads compete for the same dominant pattern during training; later, the learning dynamics transition from competitive to cooperative, leading to functional specialization across heads.

**Strengths:**

* The figures are thoughtfully designed and effectively convey the key results.
* The theoretical development is clear and satisfying: it provides a principled dynamical systems view of an empirically observed phenomenon.
* The connection between optimization dynamics and low-rank tensor factorization is elegant and offers intuitive insight into head specialization.

**Weaknesses:**

1. Missing Discussion of Generalization.  The abstract mentions “generalization,” yet this topic is not revisited in the main text.
2. Limited Scope of Contribution. The first stated contribution—analyzing a simplified, single-layer model—should be reframed as a limitation rather than a contribution. It remains unclear how the conclusions would extend to deeper or larger-scale architectures or more generalized type of sequences.
3. Unclear Cognitive Relevance. The setup—modeling the next token as a weighted superposition of prior chunks of tokens—deviates substantially from linguistic or cognitive formulation of hierarchical structures in sequences. Language is not well approximated by a linear summation of token embeddings.
    * How canonical is this formulation for studying sequences in general?
    * How does it compare to more structured formulations of sequences (e.g., Wu et al., NeurIPS 2022; Wu et al., ICLR 2025)?

**Questions:**

1. Copying vs. Reuse.  Lines 38–41 mention compositionality and the reuse of sub-components. Please clarify the distinction between copying, reusing, and re-binding (cf. Wu 2022, Wu 2024 ICLR) and how these mechanisms relate to the learning dynamics observed here.
2. Conceptual Connection to Prior Work.  The relation to recent work such as Zucchet et al. on compositional knowledge formation should be discussed. How does the present theory complement or differ from these accounts?
3. Interpretation of Results. In Fig. 3, why does the KL divergence for A A1∗ increase after the first training stage, given that it measures the divergence between the ground truth and a transformer with unrestricted context length?


Reference:

_Zucchet, N., D’Angelo, F., Lampinen, A. K., & Chan, S. C. Y. (2025). The emergence of sparse attention: impact of data distribution and benefits of repetition. arXiv preprint arXiv:2505.17863._

_Wu, S., Thalmann, M., Dayan, P., Akata, Z., & Schulz, E. (2025). Building, Reusing, and Generalizing Abstract Representations from Concrete Sequences. The Thirteenth International Conference on Learning Representations (ICLR 2025)._

_Wu, S., Élteto, N.,Dasgupta, I., & Schulz, E. (2022). Learning Structure from the Ground-up—Hierarchical Representation Learning by Chunking. 36th Conference on Neural Information Processing Systems (NeurIPS 2022)._

---

> ### Author Response · Authors · 2025-11-21
>
> We thank the reviewer for insightful questions. Below, we provide answers to listed weaknesses and questions.
>
>
> > 1. Missing Discussion of Generalization.  The abstract mentions “generalization,” yet this topic is not revisited in the main text.
>
> We provide a discussion on generalization in Section 2.5. where we demonstrate that the number of bumps in the loss curves depend on the dataset size.
>
> > 2. Limited Scope of Contribution. The first stated contribution—analyzing a simplified, single-layer model—should be reframed as a limitation rather than a contribution. It remains unclear how the conclusions would extend to deeper or larger-scale architectures or more generalized type of sequences.
>
> We **extended our empirical settings** to two-block architectures and various augmented sequences in Appendix B. In particular, we studied the effect of depth in transformers, non-uniform attention weights within intervals or overlapping intervals.
>
> Our analysis pertains only to simplified single-layer. Such idealized settings are standard in the literature for theoretical anaylsis [1,4,5].
>
> > 3. Unclear Cognitive Relevance. The setup—modeling the next token as a weighted superposition of prior chunks of tokens—deviates substantially from linguistic or cognitive formulation of hierarchical structures in sequences. Language is not well approximated by a linear summation of token embeddings.
>     * How canonical is this formulation for studying sequences in general?
>     * How does it compare to more structured formulations of sequences (e.g., Wu et al., NeurIPS 2022; Wu et al., ICLR 2025)?
>
> These are excellent questions. We agree that our synthetic sequences do not have any hierarchical structure. This is by design as we wanted to study the simplest setting in which the phenomena appears.
>
> Overall, simple synthetic sequences have been key in understanding transformers [1-5]. Sequences in our work does not have any depth structure. That is, they are directly on atomic units without any chunking.
>
> We cited the papers raised by the reviewer in the related work section to highlight that n-gram models without specifical structural assumptions lack the hierarchical linguistic features in natural language.
>
> [1] Nichani, Eshaan, Alex Damian, and Jason D. Lee. "How transformers learn causal structure with gradient descent." arXiv preprint arXiv:2402.14735 (2024).
>
> [2] D'Angelo, Francesco, Francesco Croce, and Nicolas Flammarion. "Selective induction heads: How transformers select causal structures in context." The Thirteenth International Conference on Learning Representations. 2025.
>
> [3] Varre, Aditya, Gizem Yüce, and Nicolas Flammarion. "Learning In-context n-grams with Transformers: Sub-n-grams Are Near-stationary Points." arXiv preprint arXiv:2508.12837 (2025).
>
> [4] Marion, Pierre, et al. "Attention layers provably solve single-location regression." arXiv preprint arXiv:2410.01537 (2024).
>
> [5] Duranthon, Odilon, et al. "Statistical advantage of softmax attention: insights from single-location regression." arXiv preprint arXiv:2509.21936 (2025).

---

> > ### Author Response · Authors · 2025-11-21
> >
> > > 1. Copying vs. Reuse.  Lines 38–41 mention compositionality and the reuse of sub-components. Please clarify the distinction between copying, reusing, and re-binding (cf. Wu 2022, Wu 2024 ICLR) and how these mechanisms relate to the learning dynamics observed here.
> >
> > In our paper, **copying** refers to the computational act of copying data from one part of the network to another. This is a lower-level computational operaration and orthogonal to abstract representations that are the study of these papers. As we discussed, our task requires no abstraction as the sequences do not have any hierarchical structure.
> >
> > Note that the **copying** primitive is required for chunk identification. A transformer model that needs to identify a chunk of certain length need to pass the information of the previous tokens to others in order to recognize the chunk. This is achieved via copying.
> >
> > > 2. Conceptual Connection to Prior Work.  The relation to recent work such as Zucchet et al. on compositional knowledge formation should be discussed. How does the present theory complement or differ from these accounts?
> >
> > Zucchet et al. study a setting that corresponds $h=1$ in our paper. They are interested in how fast one escapes the initialization $(V = 0, s = \frac{1}{T} 1_T)$ and the effect of burstiness/repetition on the escape time. The main technique of analysis is local Taylor approximation around this origin. Notably, the origin is not a saddle.
> >
> > Our theoretical analysis is on the whole dynamics that follows after this initial escape. When $h > 1$, this leads to incremental learning with saddle-to-saddle dynamics. We establish the convergence of the dynamics to the first saddle when heads are coupled. In Appendix D, **we added a new analysis** of convergence to other saddles by assuming certain exiting points from the previous saddles. Here, we need to rely on certain invariances of the trajectories.
> >
> > We added this comparison to the related work section.
> >
> > > 3. Interpretation of Results. In Fig. 3, why does the KL divergence for A A1∗ increase after the first training stage, given that it measures the divergence between the ground truth and a transformer with unrestricted context length?
> >
> > Figure 3 measures the divergences between functions defined in Equation 3 and the transformer. As the transformer incorporates the second feature matrix $A_2^\star$, it deviates from predictions with just using the first feature matrix, $A_1^\star$.

---

> > > ### Author Response · Authors · 2025-11-27
> > >
> > > Dear Reviewer dRyx,
> > >
> > > As the ICLR discussion period is closing soon, we wanted to kindly remind you that any additional feedback or clarifications you may have would be greatly appreciated. Your insights are extremely valuable and help us strengthen the paper during this stage.
> > >
> > > Thank you again for your time and thoughtful engagement.

---

### Official Review · Reviewer_4xLi · 2025-11-01

**Soundness:** 3
**Presentation:** 2
**Contribution:** 2
**Rating:** 4
**Confidence:** 3

**Summary:**

This paper investigates how a single-layer transformer learns sparse attention patterns for next-token prediction in a synthetic task where tokens depend on non-overlapping blocks of past tokens with varying importance. Through experiments, the authors observe an incremental learning dynamic: the model first enters a competitive phase where all attention heads focus on the highest-norm block, followed by a cooperative phase where heads gradually specialize to attend to lower-norm blocks. They support this finding with a theoretical analysis that reduces the training dynamics to a tensor factorization problem, proving convergence properties for the competitive phase. Additional experiments explore the impact of dataset size, initialization scale, and block importance hierarchy on generalization.

**Strengths:**

1. The synthetic task cleanly isolates how transformers learn hierarchical sparse dependencies, providing a tractable testbed for analyzing training dynamics.

2. The theoretical analysis offers a principled explanation for the competitive phase, complementing empirical observations.

3. Experiments on dataset size reveal that limited data induces implicit regularization(learning fewer blocks), deepening understanding of generalization in data-scarce regimes.

**Weaknesses:**

1. While previous works like [1] have studied how transformers learn causal structure, this paper provides a more detailed analysis of the training dynamics; however, most of the analysis remains experimental, and the finding that the model first learns to attend to the most important tokens and then refines the pattern seems intuitive and not surprising.

2. The theoretical analysis lacks a clear explanation; for example, V(t) and s(t) are not defined, making it hard to grasp the main statement of each theorem.

3. Conclusions are derived from an idealized setup (single-layer transformer, synthetic Markov data). It remains unclear if dynamics hold in deeper architectures, with natural data, or under standard techniques like layer normalization.

[1] Nichani, Eshaan, Alex Damian, and Jason D. Lee. "How transformers learn causal structure with gradient descent." arXiv preprint arXiv:2402.14735 (2024).

**Questions:**

Please refer to the weakness part. If there are any misunderstandings on my part, please point them out, and I will reconsider my evaluation of this work.

---

> ### Author Response · Authors · 2025-11-21
>
> We thank the reviewer for the detailed and actionable feedback on the analysis and clarity. Below, we address these issues and point out our improvements based on the manuscript.
>
> > While previous works like [1] have studied how transformers learn causal structure, this paper provides a more detailed analysis of the training dynamics; however, most of the analysis remains experimental, and the finding that the model first learns to attend to the most important tokens and then refines the pattern seems intuitive and not surprising.
>
> We added Appendix D, where we **substantially extend our analysis**. We note that obtaining an exact characterization of the training dynamics is very challenging, as even tensor factorization remains an active area of research. Our work provides a novel analysis of attention that explains the individual dynamics at each stage. This requires new techniques that control how attention dynamics maintain overall alignment. For these reasons, we believe our work offers strong theoretical contributions that explain the empirical phenomena studied in the paper.
>
> We agree that incremental learning may not be surprising, but it remains important because stage-wise dynamics have significant implications for generalization and emergence. We introduce a theory-friendly setting in which we provide a sharp mathematical characterization, consistent with recent works such as [1,2].
>
> > The theoretical analysis lacks a clear explanation; for example, V(t) and s(t) are not defined, making it hard to grasp the main statement of each theorem.
>
> When $(V_i(0), s_i(0))$ and $(V_j(0), s_j(0))$ are initialized the same, they coevolve, i.e., $V_i(t) = V_j(t), s_i(t) = s_j(t)$, as the gradient is the same. Now, when all of the heads are initialized the same, we obtain $V_i(t) = V(t), s_i(t) = s(t)$ for some $V$ and $s$. We improved the presentation to make this point more clear.
>
> > Conclusions are derived from an idealized setup (single-layer transformer, synthetic Markov data). It remains unclear if dynamics hold in deeper architectures, with natural data, or under standard techniques like layer normalization.
>
> We provide experiments that include a single full-stack transformer (including layer normalization and MLPs) in Section 2. We also add **new experiments with two-block transformers** in Section B.5. We observe that the incremental learning behavior is still present but less pronounced for the first phase in two-block transformers. This is similar to findings of [6].
>
> These idealized setting (data, architecture) are such that the phenomena of study is isolated which is standard in the literature [3-7].
>
> [1] Duranthon, Odilon, et al. "Statistical advantage of softmax attention: insights from single-location regression." arXiv preprint arXiv:2509.21936 (2025).
>
> [2] Zucchet, Nicolas, et al. "The emergence of sparse attention: impact of data distribution and benefits of repetition.". 39th Conference on Neural Information Processing Systems (NeurIPS 2025).
>
> [3] Nichani, Eshaan, Alex Damian, and Jason D. Lee. "How transformers learn causal structure with gradient descent." arXiv preprint arXiv:2402.14735 (2024).
>
> [4] Edelman, Ezra, et al. "The evolution of statistical induction heads: In-context learning markov chains." Advances in neural information processing systems 37 (2024): 64273-64311.
>
> [5] D'Angelo, Francesco, Francesco Croce, and Nicolas Flammarion. "Selective induction heads: How transformers select causal structures in context." The Thirteenth International Conference on Learning Representations. 2025.
>
> [6] Varre, Aditya, Gizem Yüce, and Nicolas Flammarion. "Learning In-context n-grams with Transformers: Sub-n-grams Are Near-stationary Points." arXiv preprint arXiv:2508.12837 (2025).
>
> [7] Zhang, Y., Singh, A.K., Latham, P.E. & Saxe, A.M. "Training Dynamics of In-Context Learning in Linear Attention." Proceedings of the 42nd International Conference on Machine Learning (2025).

---

> > ### Author Response · Authors · 2025-11-27
> >
> > Dear Reviewer 4xLi,
> >
> > As the ICLR discussion period is closing soon, we wanted to kindly remind you that any additional feedback or clarifications you may have would be greatly appreciated. Your insights are extremely valuable and help us strengthen the paper during this stage.
> >
> > Thank you again for your time and thoughtful engagement.

---

### Official Review · Reviewer_yNiz · 2025-11-01

**Soundness:** 3
**Presentation:** 2
**Contribution:** 2
**Rating:** 4
**Confidence:** 4

**Summary:**

The paper introduces a synthetic task to study how transformers incrementally learn features of varying importance during training. The task is inspired by n-gram or Markov-chain settings, but the relative importance of previous tokens is explicitly controlled through feature/weight matrices with different norms.

Empirically, the authors train transformers on this task and show that the models learn in stages: first capturing the most important features or positions, then gradually incorporating less important ones. During this process, attention heads initially all compete to represent the most important features, but later each specializes in distinct subsets of positions according to their importance. The paper also includes ablation studies analyzing the effects of initialization scale, the feature scale, and dataset size.

Theoretically, the authors analyze a regression variant of the task trained with MSE loss with a simplified architecture. They study the gradient flow dynamics and show that, under symmetric initialization, all heads focus on the most important positions, whereas small deviations from symmetry lead different heads to take different paths.

**Strengths:**

The paper is well-motivated, and the problem setup is clearly presented. The task design is well-suited for studying learning dynamics in a simplified and controlled testbed that also lends itself to theoretical analysis (though the theoretical part later relies on additional simplifications).

For the proposed task, the empirical study effectively reveals and visualizes the inner mechanisms and learning dynamics, offering nice insight into how attention heads specialize during training.

The overall presentation is clear and well-structured, with some exceptions discussed later.

**Weaknesses:**

- The theoretical analysis is conducted on a much simpler setup than the synthetic task used in the empirical study. This level of simplification is not inherently problematic for a theoretical treatment, provided the simplified model replicates the key behaviors and offers a tractable framework for analysis. However, the theory presented here is only partial: it focuses solely on the initial stage of training, where all heads learn the same pattern under specific initialization assumptions. In addition, the presentation of the theory section could be improved for clarity (see questions below).

- The paper could more clearly situate its findings in relation to prior work. For instance, the Markov-chain setup of Edelman et al. (2024) can be viewed as a special case of the proposed task (e.g., with $h=1$ and all $\alpha$ values equal), where all positions have the same importance. Yet even in that setting, stage-wise learning progression is observed. Here, however, the stage-wise progression appears mainly due to $h>1, m>1$ (the difference in positional importance), and the positions with the same importance, say all positions in $I(1)$, seem to be learned simultaneously and not stage-wise anymore.

    There are also two more papers that study similar synthetic task studies for training dynamics, which are relevant to cite.

    [1] Zhou et al. “Transformers Learn Variable-Order Markov Chains In-Context.”

    [2] Deora et al. “In-Context Occam’s Razor: How Transformers Prefer Simpler Hypotheses on the Fly.”

**Questions:**

1. In all experiments, the $\alpha$ values are chosen uniformly within each group $I(k)$. What would be the impact of using non-uniform $\alpha$ on the training dynamics? Would that also contribute to the stage-wise learning behavior?

2. In Figure 4 (left), what is the baseline curve?

3. The notation in Section 3 is a bit confusing. (i) Is $d_i$ (line 378) defined? (ii) Since $V_k$ and $s_k$ are trainable parameters, shouldn’t they carry a time index $t$ in Theorem 2 (first equation) and Theorem 3?

4. Could you clarify Figure 10? What do the three plotted features and positions represent?

5. In Theorem 1, does the fact that $m_1*$ is the largest (specifying the most important position) play a role in the proof, or is the alignment of the fixed point with $V_1^*$ solely a result of the initialization assumption (6)?

6. Regarding Section 3.3, how does the theoretical setup demonstrate the cooperation phase? Theorem 3 still shows that all heads remain close to each other for a finite time, but there doesn’t seem to be a result describing what happens afterward, is that correct?

*Minor:*
- In Section 2.2, it would be helpful to mention that $w=12$ in the main text (currently stated only in the Appendix) so that the context lengths $c$ in paragraph 186 are clearer.
- In line 195, should it say **un**restricted context length?

---

> ### Author Response · Authors · 2025-11-21
>
> We thank the reviewer for the detailed review, which allowed us to significantly improve the manuscript. Below, we answer to each weakness and questions, pointing out the improvements in the manuscript.
>
> > However, the theory presented here is only partial: it focuses solely on the initial stage of training, where all heads learn the same pattern under specific initialization assumptions.
>
> We **extended our theory to the collaborative phases in Appendix D**. This is done by analyzing initializations where the first $n$ heads are learned and there is a single head that is breaking away from the "ensemble of heads" that focus on the first position. We show that sequential learning proceeds collaboratively: the fast dynamics of the ensemble steer the offshoot head along a monotonic trajectory until the $(n+1)$-st position is learned.
>
> We remark that the original matrix/tensor factorization problem is still open. Our problem is harder for two for reasons: (i) the saddles in our problems are more intricate, (ii) we deal with the softmax nonlinearity. Thus, the complete characterization of the dynamics is elusive.
>
> > 6. Regarding Section 3.3, how does the theoretical setup demonstrate the cooperation phase? Theorem 3 still shows that all heads remain close to each other for a finite time, but there doesn’t seem to be a result describing what happens afterward, is that correct?
>
> Your understanding is correct. We extended our theory to the collaborative phases after your review. These main theorems from these results are added to Section 3.3.
>
> > The paper could more clearly situate its findings in relation to prior work. For instance, the Markov-chain setup of Edelman et al. (2024) can be viewed as a special case of the proposed task ...
>
> We first commment relationship of our work with Edelman et al. (2024) and then comment on our main contributions with respect to prior work. Finally, we clarify the purpose of the $m > 1$ assumption.
>
> **Comparison with Edelman et al. (2024)**: Edelman et al. (2024) consider **in-context** Markov chain tasks, where each sequence is drawn from a freshly sampled Markov chain. The solution of this task is the counting estimator. Instead, we consider learning from a **single** Markov chain which is substantially simpler since we only have to learn a single transition kernel. Therefore, these two tasks are statistically different and not related even for $h=1$.
>
> **Our work**: Our main contribution is to isolate the interplay of sparsity in attention patterns and learning dynamics as the driving force for incremental learning. We claim that this holds regardless of the data setting and the simplest instantiation (in terms of model and data complexity) is the one we provide. This is an effort to simplify in order to get to the essence of the phenomenon. A perk of this simplification is that it leads to a clean and accessible theory, which is closely related to matrix/tensor factorization.
>
> This is in contrast with in-context settings, where one needs to study more complicated data processes. As shown in Varre et al. (2025), even the characterization of the saddles is very difficult and requires a limit over the sequence length.
>
> **The use of $m > 1$**: In principle, there are many sources of stage-wise dynamics. For example, in $h=1$ case, $A_1^\star$ can have features of different norms which could lead to different speeds of learning and incremental learning depending on the initialization. Or, the order of learning might be influenced by the biases of the architecture instead of just statistical importances. This is highlighted in Varre et al. (2025) which explains the "usual" order of learning in Edelman et al. (2024).
>
> When $m=1$, we still observe incremental learning (see Figure 4) but it is less stable. Overall, the assumption of $m > 1$ captures features with different statistical importances. This is the key element that determines how large an initialization basin will exhibit stage-wise behaviors. In the theoretical limit of certain initializations, we still expect to see stage-wise dynamics regardless of the value of $m$. However, our results are not for a particular limit of initialization but for a large initialization set which is stronger. We comment more on the initialization in Appendix E.
>
> > There are also two more papers that study similar synthetic task studies for training dynamics, which are relevant to cite.
>
> We thank the reviewer for bringing these papers to our attention. We added these papers to the manuscript.
>
> > 1. In all experiments, the $\alpha$ values are chosen uniformly within each group $I(k)$. What would be the impact of using non-uniform on the training dynamics? Would that also contribute to the stage-wise learning behavior?
>
> We added new experiments in Appendix B.6 to address non-uniform $\alpha$ values. Qualitatively, we do not observe any significant changes from uniform $\alpha$ values.

---

> > ### Author Response · Authors · 2025-11-21
> >
> > > 2. In Figure 4 (left), what is the baseline curve?
> >
> > It is the run that corresponds to initialization scale of $1$. This is from the run used in Section 2.2, which we referred to as "baseline". In order to increase the clarity, we will explicitly write it as $1$.
> >
> > > 3. The notation in Section 3 is a bit confusing. (i) Is $d_i$ (line 378) defined? (ii) Since $V_k$ and $s_k$ are trainable parameters, shouldn’t they carry a time index $t$ in Theorem 2 (first equation) and Theorem 3?
> >
> > Thanks for catching this typos.
> >
> > (i) We meant to write $V_i = 0$.
> >
> > (ii) $V_k$ and $s_k$ in these equations should indeed have time indices.
> >
> > > 4. Could you clarify Figure 10? What do the three plotted features and positions represent?
> >
> > Position (1), (2), (3) refers to the projections of the attention scores $s_1, s_2, s_3$ to $s_1^\star, s_2^\star, s_3^\star$, respectively. Similarly, Feature (1), (2), (3) refers to the projections of the value matrices $V_1, V_2, V_3$ to $V_1^\star, V_2^\star, V_3^\star$, respectively. The last subplot shows the individual contributions of $(V_1^\star, s_1^\star), (V_2^\star, s_2^\star), (V_3^\star, s_3^\star)$ to the loss.
> >
> > > 5. In Theorem 1, does the fact that $m_1^\star$ is the largest (specifying the most important position) play a role in the proof, or is the alignment of the fixed point with $V_1^\star$ solely a result of the initialization assumption (6)?
> >
> > It does play a crucial role. This is the key reason why any initialization with such alignment converges to the same point. We added a new section Appendix E that discusses how to expand this initialization scheme.
> >
> > > In Section 2.2, it would be helpful to mention that $w=12$ in the main text (currently stated only in the Appendix) so that the context lengths  in paragraph 186 are clearer.
> >
> > > In line 195, should it say unrestricted context length?
> >
> > Thanks for these comments. We improved the manuscript to make these two points more clear.

---

> > > ### Author Response · Authors · 2025-11-27
> > >
> > > Dear Reviewer yNiz,
> > >
> > > As the ICLR discussion period is closing soon, we wanted to kindly remind you that any additional feedback or clarifications you may have would be greatly appreciated. Your insights are extremely valuable and help us strengthen the paper during this stage.
> > >
> > > Thank you again for your time and thoughtful engagement.

---

### Author Response · Authors · 2025-12-01

Dear Area Chairs and Reviewers,

Thank you for your time and effort during the review and rebuttal process, especially given the unusual situation. We truly appreciate your work in keeping the evaluation fair and constructive.

We conclude our rebuttal with a summary of the main changes made in response to the reviewers’ comments:

(i) We substantially extended our theoretical results to address the concerns from Reviewer yNiz and Reviewer 4xLi. In particular, our theory now covers collaborative phases of the dynamics in addition to the initial competitive phase, and we clarified how our results improve upon prior work.

(ii) We expanded our experimental analysis to cover all requested scenarios, including non-uniform attention weights, overlapping intervals, alternative optimizers, and different depths. We observe similar incremental learning behavior under these new scenarios.

(iii) We clarified our contributions relative to works such as Edelman et al. (2024), Zucchet et al. (2025), and Wu et al. (2022), emphasizing that we study copying circuits that arise in practice in deep architectures, using a simplified synthetic setting to better isolate and theoretically characterize their dynamics.

We believe we have addressed all of the reviewers’ concerns and sincerely appreciate your consideration.

Warm regards,
Authors of Submission 2407

---

### Meta-Review · Area_Chair_zGwm · 2026-01-08

**Summary:**

This submission studies the incremental learning dynamics of a shallow multi-head transformer in learning a high-order Markov chain. The reviewers raised the following concerns.

* Reviewers ShLm and dRyx mentioned the limited (synthetic) scope of the problem setting and strong assumptions in the analysis. The authors partly addressed this concern by including additional experiments in more realistic settings, and suggesting that similar simplification is standard in prior theoretical works.

* Reviewers 4xLi and yNiz complained about the gap between the synthetic task used in the empirical study and the theoretical setting. The authors partly addressed this concern by expanding their theoretical analysis to explain additional phases in the learning dynamics.

* Reviewers yNiz, ShLm, and dRyx requested the authors discuss overlap with prior works by Edelman et al. and Zucchet et al., which the authors explained in the rebuttal.

Overall, the authors addressed some of the major concerns in the discussion period; however, the added theoretical content has not been thoroughly peer reviewed, and the area chair believes that the submission would benefit from another round of revision.

**Reviewer Concerns:**

See above.

**Reviewer Scores:**

Reviewers yNiz, 4xLi, and ShLm all gave a score of borderline reject. While their concerns are partly addressed, reviewer ShLm was inclined to keep the original evaluation, whereas the other two reviewers did not indicate a change in their opinion. The area chair believes that these 3 reviewers are likely to maintain their borderline scores.

---

### Decision · Program_Chairs · 2026-01-26

Reject